# Sea surface temperature evolution of the North Atlantic Ocean across the Eocene-Oligocene Transition

Kasia K. Śliwińska[1,2,*], Helen K. Coxall[3,4], David K. Hutchinson[3,5], Diederik Liebrand[6], Stefan Schouten[2,7], Agatha M. de Boer[3,4]

[1]Department of Geoenergy and storage, Geological Survey of Denmark and Greenland (GEUS), Øster Voldgade 10, 1350 Copenhagen, Denmark

[2]NIOZ Royal Netherlands Institute for Sea Research, Department of Marine Microbiology and Biogeochemistry, Landsdiep 4, 1797 SZ 't Horntje, Texel, the Netherlands

[3]Department of Geological Sciences, Stockholm University, Svante Arrhenius väg 8, 114 18 Stockholm, Sweden

[4] Bolin Centre for Climate Research, Stockholm University, Stockholm, Sweden

[5]Climate Change Research Centre, University of New South Wales, Sydney NSW 2052, Australia

[6]National Oceanography Centre, European Way, SO14 3ZH, Southampton, United Kingdom

[7]Department of Earth Sciences, Faculty of Geosciences, Utrecht University, Vening Meinesz building A, Princetonlaan 8a, 3584 CB Utrecht, the Netherlands

*Correspondence to*: Kasia K. Śliwińska (kksl@geus.dk) and Agatha M. de Boer (agatha.deboer@geo.su.se)

**Key words:** sea surface temperature evolution, Eocene Oligocene Transition, Atlantic Ocean, coupled climate model, ODP Site 647, southern Labrador Sea

**Abstract.** A major step in the long-term Cenozoic evolution toward a glacially-driven climate occurred at the Eocene Oligocene Transition (EOT), ~34.44 to 33.65 million years ago (Ma). Evidence for high latitude cooling and increased latitudinal temperature gradients across the EOT has been found in a range of marine and terrestrial environments. However, the timing and magnitude of temperature change in the North Atlantic remains highly unconstrained. Here, we use two independent organic geochemical paleo-thermometers to reconstruct sea surface temperatures (SSTs) from the southern Labrador Sea (Ocean Drilling Program - ODP Site 647) across the EOT. The new SST records, now the most detailed for the North Atlantic through the 1 million years leading up to the EOT onset, reveal a distinctive cooling step of ~3 °C (from 27 to 24 °C), between 34.9 Ma and 34.3 Ma, which is ~500 kyr prior to Antarctic glaciation. This cooling step, when compared visually to other SST records is asynchronous across Atlantic sites, signifying considerable spatiotemporal variability in regional SST evolution. However, overall, it fits within a phase of general SST cooling recorded across sites in the North Atlantic in the 5 million years bracketing the EOT.

Such cooling might be unexpected in light of proxy and modelling studies suggesting the start-up of the Atlantic Meridional Overturning Circulation (AMOC) before the EOT, which should warm the North Atlantic. Results of an EOT modelling study (GFDL CM2.1) helps reconcile this, finding that a reduction in atmospheric $CO_2$ from 800 to 400 ppm may be enough to counter the warming from an AMOC start-up, here simulated through Arctic-Atlantic gateway closure. While the model simulations applied here are not yet in full equilibrium and the experiments are idealised, the results, together with the proxy data, highlight the heterogeneity of basin-scale surface ocean responses to the EOT thermohaline changes, with sharp temperature contrasts expected across the northern North Atlantic as positions of the subtropical and subpolar gyre systems shift. Suggested future work includes increasing spatial coverage and resolution of regional SST proxy records across the North Atlantic to identify likely thermohaline fingerprints of the EOT AMOC start-up, as well as critical analysis of the causes of inter-model responses to help better understand the driving mechanisms.

## 1. Introduction

The principal signature of climatic change across the EOT in deep marine records is an apparent two-step positive increase in the oxygen isotopic ($\delta^{18}$O) composition of deep sea foraminifera, centred around 34 Ma (Zachos et al., 1996; Coxall et al., 2005) (Supplementary Information). Current understanding is that the first $\delta^{18}$O step mostly reflects ocean cooling (Step 1; 33.9 Ma, known previously as EOT-1; see Hutchinson et al., 2021) and the second step reflects the accumulation of terrestrial ice on Antarctica (Lear et al., 2008; Bohaty et al., 2012; Zachos et al., 1996), which was recently redefined as the Early Oligocene oxygen Isotope Step (EOIS); at around 33.6 Ma (Hutchinson et al., 2021) (see also Supplementary Information). While a cooling signal is recorded in the benthic realm, its absolute amplitude, expression at the surface ocean, and its global extent and uniformity remain largely unconstrained. A variety of data types support the EOT cooling in the low latitudes and the Southern high latitudes, revealing temperature decreases that range between 2.5 to 5 °C in the deep sea (Bohaty et al., 2012; Lear et al., 2008; Pusz et al., 2011) and between 2 to 6 ºC in surface waters and on land (Bohaty et al., 2012; Haiblen et al., 2019; Lauretano et al., 2021; Liu et al., 2009; Tibbett et al., 2021; Wade et al., 2012). Temperature evolution of the high northern latitudes, including regions of the North Atlantic Ocean where deep water is formed in the present day (Broeker, 1991; de Boer et al., 2008), however, remains less documented.

Existing low-resolution paleoclimate reconstructions from the Norwegian–Greenland Sea, including SST and terrestrial temperature constraints from palynology (Eldrett et al., 2009), organic molecular fossils (Liu et al., 2009; Schouten et al., 2008), and sediment grains (i.e., ice-rafted debris) (e.g. Eldrett et al., 2007), suggest some degree of cooling and increased seasonality concurrent with the EOT, which is possibly tied to relatively minor land-ice expansions on Greenland (Eldrett et al., 2007; Bernard et al., 2016). Records from the mid-latitude North Atlantic report no SST change across the EOT as evidence of a temporary decoupling of the North Atlantic Ocean from the southern high latitudes and thus hemispherical asymmetric cooling, attributed to changes in circulation-driven heat transport (Liu et al., 2018). This existing suite of northern EOT temperature records, still provide sparse coverage, with gaps at critical stages in the late Eocene and are of generally low temporal resolution, especially in the 1 million year-lead interval prior to the EOT onset. These data can therefore not be correlated in great detail to the EOT as identified in global benthic foraminiferal $\delta^{18}$O records. This limits the

understanding of cause-and-effect relationships with the much better resolved $\delta^{18}O$ and deep sea temperature records from the Southern Hemisphere (e.g. Hutchinson et al., 2021). A further stumbling block is that the quality of many northern North Atlantic records is often compromised by (*i*) carbonate dissolution in the sub-Arctic North Atlantic, that limits proxy based temperature estimates using foraminiferal calcite, and (*ii*) gaps in the sedimentary record at many sites across the Eocene/Oligocene boundary that are caused by deep sea erosion linked to bottom water current strengthening (e.g. Miller et al., 1985).

Here we present new proxy records of sea surface temperature from ODP Site 647 (53°20'N 45°16'W), located in the western North Atlantic (Fig. 1), across an upper Eocene to middle Oligocene (i.e., time equivalent to ~38–~26.5 Ma) succession of hemipelagic clay from the southern Labrador Sea. We use the $TEX_{86}$ and $U_{37}^{KI}$ proxies (Fig. 2), which are two independent paleothermometers based on fossil organic biomarkers derived from archaea and photosynthetic plankton, respectively (Schouten et al., 2002; Brassell et al., 1986). These new data constitute the best-resolved EOT-spanning SST proxy records from the Northern hemisphere to date. They document patterns of temperature change in the north western Atlantic and help decipher the complex temperature evolution of the (North) Atlantic Ocean across the largest climate state-change of the Cenozoic Era.

We compare our newly obtained SST record to published SST proxy records and reconstruct latitudinal SST gradients for the Eocene and Oligocene in the North Atlantic (Fig. 3 and 4). The compilation of SST records (Fig. 3a) show cooling in the Atlantic across the EOT that one might expect as part of the global transitioning into an icehouse world and which is usually attributed to a reduction in atmospheric $CO_2$ (Anagnostou et al., 2016; Cramwinckel et al., 2018). Hypotheses for the $CO_2$ decrease abound and include gradually reduction of tectonically driven outgassing, expansion of marine carbon sinks (Müller et al., 2022), weathering or biological pump feedbacks from an AMOC start-up (Hutchinson et al., 2021; Elsworth et al., 2017; Fyke et al., 2015). The AMOC has been suggested by multiple proxies to become active around the time of the EOT (Borrelli et al., 2021, 2014; Boyle et al., 2017; Coxall et al., 2018; Hutchinson et al., 2019; Kaminski and Ortiz, 2014,; Langton et al., 2016; Uenzelmann-Neben and Gruetzner, 2018; Via and Thomas, 2006). Theory and modelling work have attributed the AMOC start-up alternatively to Arctic closure (Hutchinson et al., 2019; Straume et al., 2022) the deepening of Drake Passage and/or the Tasman gateway (Toggweiler and Bjornsson, 2000), and the deepening of the Greenland Scotland Ridge (Stärz et al., 2017). A main feature of the AMOC is its northward heat transport in the Atlantic, which acts to warm the high latitude North Atlantic more than it would be otherwise expected, begging the question of how AMOC warming and $CO_2$ cooling may combine to produce reconstructed cooling in the North Atlantic. To address this question, we here analyse the SST patterns in the modelling output from Hutchinson et al. (2018, 2019) (model GFDL CM2.1), in which they compared the impact of Arctic closure (causing Atlantic salinification sufficient to trigger deep sinking) and an atmospheric $CO_2$ decrease on the deep ocean circulation. They concluded that only the Arctic closure could lead to a start-up of the AMOC at the EOT (other mechanisms failed to initiate AMOC sinking). This finding was corroborated recently by Straume et al. (2022), even though the authors closed off the Arctic-Atlantic connection via different tectonic changes than Hutchinson et al. (2019). Here we focus on the implications of these processes on SST.

The manuscript starts with a description of the drilling site and core, followed by detail on the various data methods used in the study and a description of the model and simulations. The results address first the specific SST time series in the Site 647 record and then analyses the new dataset in the context of available North Atlantic SST records. The data are then compared to the modelling simulations and the implications for the processes in the North Atlantic and at the core site is discussed. We conclude with summary of the results and the potential implications for the state of knowledge of what happened at the EOT.

## 2. Labrador Sea Ocean Drilling Program Site 647

ODP Hole 647A constitutes the most northerly location (53°N) where a complete upper Eocene– middle Oligocene sedimentary sequence is known to be present (Coxall et al., 2018; Firth et al., 2013) (Fig. 1). The studied succession consists of greyish-green, moderately to strongly bioturbated nannofossil claystone and nannofossil chalk (see Supplementary information). The core recovery across the EOT (Cores 27R to 30R) is reasonable (Fig. S1). However, Core 29R is heavily disturbed (Fig. S1) and is usually omitted in the analysis of the site (Firth, 1989; Kaminski and Ortiz, 2014). We have processed one sample from the Core 29R (29R-4, 130-132, 275.5 mbsf - meters below see floor; Fig. S1) for calcareous nannofossils and biomarkers. In the sample we

found neither caved (younger) nor reworked (older) calcareous nannofossil taxa (J. Firth personal communication 2013), thus, despite intra-core sediment mixing the analysed biomarker signal remains stratigraphically useful, albeit producing a time-average SST signal, potentially for the whole Core C29.

The absolute ages for the studied succession are calculated up to the depth of 214.19 mbsf, where the highest occurrence of *Reticulofenestra umbilicus* (with diameter >14 μm) is observed, which provides an absolute age of 32.02 Ma at that depth (Firth et al., 2013). The uppermost part of the studied succession belongs to the NP24 (Firth, 1989) and the normal polarity magnetochron (Firth et al., 2013), suggesting that it is probably not younger than 26.5 Ma. Overall, even with some core disturbance and other minor core recovery gaps, a bio-magnetostratigraphic age model was obtained for the interval between ~38 and ~32 Ma (Figs. 2 and 3, and S2). The datums included in the age model have been converted to the GTS2012 (Vandenberghe et al., 2012) (Figs. 2,3, and S3), using tie-points proposed by Firth et al. (2018) (Supplements).

In other deep-sea sequences across the EOT, combined $\delta^{18}O$ and magnetic reversal stratigraphy has shown that high $\delta^{18}O$ values diagnostic of the EOGM $\delta^{18}O$ increase (Oi-1 of Zachos et al., 1996; Katz et al., 2008; Coxall and Wilson, 2011; 'Step 2' of Coxall et al., 2005; EOIS of Hutchinson et al., 2021) reach a peak close to the base of the magnetochron C13n, while the prior and first phase of the EOT transition ('Step-1' of Coxall et al., 2005, and 'EOT-1' of Katz et al. 2008 and Coxall and Wilson, 2011) occurs in the previous reversed polarity zone C13r, where $\delta^{18}O$ is on average 0.5-1‰ lower. A weak spot in the Firth et al. (2013) age model for site 647 occurs close to the Eocene/Oligocene boundary due to the particularly discontinuous coring at that level (Fig. S1). Firth et al., (2013) used a depth of 270.93 mbsf as the age tie-point for the C13r/C13n reversal boundary at Site 647. Due to the sampling limits of the paleomagnetic analysis (Core 29R also exhibits sediment disturbance eliminating any coherent P-mag signal) there is a +/- 9 m uncertainty associated with this horizon (see Table S2 in Coxall et al. 2018). Our benthic $\delta^{18}O$ sample from 269.79 mbsf falls within the zone of P-mag uncertainty. Since it has a relatively low value of $\delta^{18}O$ we interpret this to be 'pre-EOGM' and (therefore pre- C13n value), thus it most likely occurs within C13r. We can therefore shift the C13r/C13n reversal depth up to 265 mbsf, which is a revised estimate of the P-mag reversal position after Firth et al., (2013).

## 3. Methods

### 3.1. Biomarkers

Organic compounds were extracted from 71 sediment samples collected from the interval between 397.60 mbsf and 135.50 mbsf ( 39R 02W 100-102cm −15R 01W 10-20cm). Samples were freeze-dried, mechanically powdered and 5-17g of sediment was taken for further analysis. The total lipid extract was obtained from sediments using the accelerated solvent extraction (ASE) technique with dichloromethane (DCM):methanol (MeOH) (9:1, v/v). Excess solvent was removed by evaporation under Nitrogen in the TurboVap®LV for 1 hour under constant temperature (30°C) and constant gas pressure (15 psi). The total lipid extract was separated over an activated $Al_2O_3$ column into apolar (hexane:DCM, 1:1, v/v), ketone (hexane:DCM, 1:1 v/v) and polar (DCM:MeOH, 1:1, v/v) fractions, respectively.

### 3.1.1. Alkenone based temperature estimates

The ketone fraction was analysed for alkenones. Sufficient concentrations of di- and tri-unsaturated alkenones were detected in 32 uppermost samples (i.e. between 241.14 to 135.50 mbsf). In these samples we calculated sea surface temperatures by applying $U_{37}^{K\prime}$ proxy (Prahl and Wakeham, 1987; Brassell et al., 1986).

First, the $U_{37}^{K\prime}$ index was calculated as follow:

$$U_{37}^{K\prime} = \frac{[C37:2]}{[C37:2]+[C37:3]} \qquad (1)$$

where the numbers in [C37:2] and [C37:3] refer to the number of carbon atom and double bonds in the molecule. Second, the index was converted into temperature following the calibration of Müller et al., (1998).

$$T = (U_{37}^{K\prime} - 0.044) / 0.033 \qquad (2)$$

The T calibration error for Eq. (2) is ±1.5°C. For seven samples, which were analysed in duplicate, the reproducibility was better than 0.6°C (Fig. S2).

Notably, the alkenones detected in our study are not originating from *Emiliania huxleyi*, a coccolithophore which has been present only for the past 270 kyr. However, as it was shown by several studies, the Paleogene ancestors show a similar response of the $U_{37}^{K'}$ index to surface temperature compared to modern day alkenone producers
(Brassell, 2014; Villanueva et al., 2002). Like any other proxies, the $U_{37}^{K'}$ index has its uncertainties, but they are generally considered to be minimal when compared to other proxies.

The calibration of Müller et al., (1998) is near identical to the culture-based calibration used for *E. huxleyi* by Prahl et al. (1988) and is commonly used to estimate the $U_{37}^{K'}$-derived SST of the late Paleogene to Neogene strata in the northern high to mid latitudes (see e.g. Liu et al., 2009; Herbert et al., 2020; Weller and Stein, 2008).


### 3.1.2. GDGT distribution

The polar fraction (containing glycerol dialkyl glycerol tetraethers; GDGTs) was concentrated under $N_2$, dissolved in hexane/isopropanol (99:1, v/v), filtered using a 0.4 µm PTFE filter and analysed using high-pressure liquid chromatography (HPLC) as described by Schouten et al., (2007). Prior to calculating the sea surface
temperatures from the TEX$_{86}$ proxy, we have evaluated the source and the distribution of GDGTs.

For detecting a methanogenic input of GDGTs we applied the %GDGT-0 index (Sinninghe Damsté et al., 2012). Studies on enrichment cultures of Thaumarcheota suggests that when %GDGT-0 values reach values above 67% the sedimentary GDGT pool may be affected by an additional (probably methanogenic) source of GDGTs. Our Eocene to Oligocene sediments has %GDGT-0 values between 26% and 63%, with a mean value of 41 %
(Supplements) and thus the GDGT pool bears no signs of methanogenic source for the sedimentary archea. Low values of the methane index (MI) (Zhang et al., 2011) and the GDGT-2/Crenarchaeol ratio (Weijers et al., 2011) (<0.25 and <0.13, respectively) exclude input of methanotrophic archaea versus Thaumarchaeota. The relative abundance of crenarchaeol isomer fCren':Cren' + Cren (O'Brien et al., 2017) in our dataset has values between 0.05 and 0.09 (Supplements) which is within the range (0.00-0.16) of values for the modern core-top sediments.
In order to eliminate samples with GDGTs which may have been influenced by non-thermal factors we calculated the Ring Index (RI) (Zhang et al., 2016). Nine samples from our data set (12.5% of all samples, n=71) are excluded from the temperature calculations due to ΔRI above |0.3| (Zhang et al., 2016) (Supplements).

Fifteen samples (21% of all samples, n=71) were excluded from the temperature calculations because too high soil- and river-derived organic matter, as suggested by the BIT index (Hopmans et al., 2004). We used a cut-off
value of 0.4 (Supplements). The BIT cut-off value for applicability of TEX$_{86}$ as SST proxy depends on the particular location, i.e. the TEX$_{86}$ value of the terrestrial GDGTs transported to the marine environment (see discussion in Schouten et al., 2013b) as well as the Mass Spectrometer settings (Schouten et al., 2013a). In the studied interval the BIT index rarely exceeds 0.35 and shows no apparent trend in time. Furthermore, for the entire sample set, we find no correlation between BIT index and TEX$_{86}$ ($R^2$=0,01).

### 3.1.3 $TEX_{86}^{H}$ and BAYSPAR based temperature estimates

Due to BIT and/or ΔRI excessing their cut-off values, 18 samples are excluded from the TEX$_{86}$ compilation (see above). Out of 71 sediment samples, 14 were analysed in duplicate and two in triplicate. In our study we have applied two calibrations for TEX$_{86}$-derived SST estimations: the $TEX_{86}^{H}$ linear calibration (Kim et al., 2010) and the TEX$_{86}$ Bayesian regression model (BAYSPAR) (Tierney and Tingley, 2014, 2015). In the modern oceans the
TEX$_{86}^{H}$ is calculated as follows:

$$TEX_{86}^{H} = \log\left(\frac{[GDGT-2]+[GDGT-3]+[Cren']}{[GDGT-1]+[GDGT-2]+[GDGT-3]+[Cren']}\right) \tag{3}$$

Raw TEX$_{86}^{H}$ values for the studied interval are between 0.56 and 0.71 with the mean value of 0.63 (1σ calibration uncertainty). SST were subsequently calculated as follows:

$$T\ [°C] = 68.4(TEX_{86}^{H}) + 38.6 \tag{4}$$

The T calibration error for Eq. (4) is ~2.5 °C. The analytical error of the SST derived from $TEX_{86}^{H}$. is ± 0.6°C.

We also calculated SST predictions using the Bayesian regression model (BAYSPAR), for which we only included the sample set as for $TEX_{86}^{H}$. We computed SSTs using the online graphical user interface located at http://bayspar.geo.arizona.edu (accessed in 2017, currently discontinued) and inserted a paleolatitude of 45°N. Spatial analogues for deep-time for the BAYSPAR were calculated with the following settings:

215        1)   Prior mean = 0.633639 (i.e. the mean $TEX_{86}$ value for the timeseries)

       2)   Search tolerance = 0.072302 (i.e. 2*STDEV.P of timeseries)

The Bayesian estimates based on the $TEX_{86}$ index values at Site 647A point to low latitude settings as modern analogues. $TEX_{86}^{H}$ and BAYSPAR calibrations show very similar paleotemperature trends. The difference in SST is between 0°C and 0.6 °C for SST above 25.6°C, and between 0.8°C and 1.9°C for SST below 25.2 °C. Overall,
the mean difference in SST is 0.8°C. SST records based on different calibrations of $TEX_{86}$ are shown in Fig. S2.

### 3.1.4. Potential bias of the $TEX_{86}$ index

Some studies suggested that $TEX_{86}$ reflect subsurface rather than surface temperatures (e.g. Lopes dos Santos et al., 2010; Huguet et al., 2007). However, the $U_{37}^{K'}$ index, which is a well-established proxy for SST, in the earliest Oligocene (covered by the interval from ~240 to ~190 mbsf) show an overall match in both absolute values and
the temperature trend as derived from $TEX_{86}^{H}$ (Figs. 2, 3, and Fig. S2). These two proxies are based on organisms with different ecological preferences and thus may reconstruct temperatures of different seasons and depths compared to each other. Nevertheless, the similarity of both records during the earliest Oligocene (covered by the interval from ~240 to ~190 mbsf) suggests that the temperatures recorded by both proxies are indicative of surface conditions. Qin et al., (2015) questioned the application of the $TEX_{86}$ proxy in sediments deposited under
low $O_2$ concentrations. However, the nature of the benthic foraminiferal assemblages (e.g. Kaminski and Ortiz, 2014), evidence of bioturbation throughout the recovered cores (Stein et al., 1989), and lack of other sedimentological features suggesting exceptionally low oxygen conditions (Eldholm et al., 1987) across the interval covering the EOT, imply that deposition took place in oxygenated bottom waters (see also Kaminski and Ortiz, 2014). There is no correlation between BIT index and $TEX_{86}$, so we can assume that $TEX_{86}$ values are
probably not biased by terrestrial input. It has been also shown that oxic degradation of biomarker lipids can affect their relative distribution and thus the $TEX_{86}$ (Huguet et al., 2009). However, we do not observe any signs of oxic degradation in the analysed material: such as a sharp increase in the BIT index values or a high degree of correlation between $TEX_{86}$ and BIT.

### 3.2. Model simulations

The simulations were performed using the coupled climate model GFDL CM2.1 (Delworth et al., 2006) adapted to late Eocene (~38 Ma) boundary conditions, as outlined in Hutchinson et al., (2018). The model uses an ocean resolution of 1° x 1.5° x 50 levels and an atmosphere resolution of 3° x 3.75° x 24 levels. The resolution of our model is in line with the most recent set of EOT climate models (e.g. Baatsen et al., 2020; Tardif et al., 2020), which allows better representation of ocean gateways than the preceding generation of EOT models. The model
was run at two end-member $CO_2$ levels of 400 and 800 ppm, and spun up for 6500 years using an iterative coupling procedure, with the last 3200 years run in fully-coupled mode (Hutchinson et al., 2018). These experiments were carried out using modern day orbital forcing parameters. In the control configuration, the palaeogeography includes shallowly open ocean gateways between the Arctic and Norwegian-Greenland Sea as likely existed for some part of the late Eocene (Lasabuda et al., 2018; Straume et al., 2020). In this configuration,
sinking occurs in the North Pacific and the Southern Ocean, but no deep-water forms in the North Atlantic. We also simulated a modified version of the model with the Arctic-Atlantic gateway fully closed, as outlined in Hutchinson et al. (2019). This change dramatically increases the salinity in the North Atlantic and enables North Atlantic deep water to form. We thus compare the mean state and response to halving $CO_2$ from 800 to 400 ppm in a configuration where there is, and where there is not an AMOC present. All simulations were run for 6500
years, using the same spin up method as applied by Hutchinson et al., (2018) except the 400 ppm Arctic closed simulation which was branched from the 800 ppm Arctic closed configuration at year 5500 and continued for 1000 model years (Fig. 5, green). The AMOC in this run is clearly not in equilibrium yet, reducing by ~10 Sv in

the last 500 years (Fig. 5b). Similarly, the SST around the area of site 647 is still decreasing by ~0.4 °C in these 500 years with no obvious reduction in this trend by year 6500, suggesting that the final state would be at least as cold as the Arctic-closed 800 ppm case (Fig. 5c, red) and potentially even colder.

It should further be noted the GFDL CM2.1 model was the only model to simulate deep sinking in the North Pacific for the DeepMIP model intercomparison project of the early Eocene (Zhang et al., 2022). Fish debris neodymium (Nd) proxy data suggest deep sinking in the Pacific, but the evidence is not conclusive yet. It is therefore currently not possible to determine which models have the most realistic ocean state (Zhang et al., 2022). Suffice to say that the models simulate a wide variety of ocean states for the same Eocene boundary conditions, so that sensitivity studies like these would also be highly model dependent.

## 4. Proxy-derived sea surface temperature

### 4.1. Sea surface temperature in the Labrador Sea

Our Site 647 TEX$_{86}$-derived record shows high and relatively stable SSTs (~27 °C) in the southern Labrador Sea from ca. 38 up to 35.5 Ma (Figs. 2, 3). Between ~35.5 and 34.9 Ma SSTs increased by ~1.5 °C. Subsequently, between ~34.9 Ma and ~34.3 Ma, SSTs decreased by ~3–4 °C, i.e. from 27 °C to 23–24 °C, depending on the TEX$_{86}$ calibration (Fig. S2; the surface water cooling is reduced by ~1°C when using $TEX_{86}^H$ calibration). Between 34.3 Ma and 33 Ma, which includes the EOT interval, SSTs remained relatively stable (Fig. 2). Long chain alkenones, on which the $U_{37}^{K\prime}$ index is based, did not appear at Site 647 before ~33 Ma (Fig. 2). This fits with the observations that alkenones in distributions similar to those of modern day producers first appeared in the global sedimentary record around the EOT is, most likely triggered by the climate driven changes (Brassell, 2014). Nevertheless, once alkenones appear at Site 647, mean SST values derived from both $U_{37}^{K\prime}$- and TEX$_{86}$ are within the same range (Fig. S1) adding confidence in the absolute temperatures that we reconstruct. Both organic proxy temperature estimates are substantially higher than present day values (5–10°C) (Fig. 2A), and in good accordance with available time-equivalent SST reconstructions for the region (Fig. 3). Overall, both paleothermometers suggest Oligocene SST (interval from ~34 Ma to ~26.5 Ma) below 26°C (Figs 2 and S2), with two temperature minima. However, with the existing uncertainties in the age model for this interval (i.e. depth from 190 to 130 mbsf; Firth et al., 2013) it is challenging to link the SST minima with the cooling episodes from the Oligocene (e.g. Wade and Pälike, 2004). This could potentially be improved by a detailed analysis of dinocysts (e.g. Śliwińska et al., 2010; Śliwińska, 2019; Śliwińska and Heilmann-Clausen, 2011), but it is outside the scope of the present study. Notably, at the older SST minimum (depth ca. 183 mbsf, Fig. S2) $U_{37}^{K\prime}$- derived SST becomes significantly colder than TEX$_{86}$-derived SST. Potentially, this may be because the surface conditions, reflected by the $U_{37}^{K\prime}$, changed more substantially than subsurface temperatures, which will affect TEX$_{86}$-derived to a larger extend. Alternatively, it could indicate that there were shifts in seasonal impacts on the proxies.

Overall, $TEX_{86}^H$ -derived SST shows a distinctive cooling step of ~3-4 °C at Site 647, when comparing the warmer Eocene (SST between 29 °C and 25.5 °C, interval from ~38 to 35.5 Ma) with the colder Oligocene (SST below 25 °C, interval from ~34 to ~26.5 Ma) (Fig. 2). Notably, most published SST data from the Atlantic Ocean (all shown in Fig. 3) are of (much) lower resolution and only bracket the main cooling and ice-growth events associated with the EOT. Our study provides the highest resolution, long term SST record from the North Atlantic region across the late Eocene, to date. It uniquely pinpoints the high northern latitude changes during the main climatic transitions and the critical lead-up period, by identifying a cooling in the southern Labrador Sea between 34.9 and 34.3 Ma, approximately 500 kyr prior to the Step 1 event (Fig 2a), possibly related with the Late Eocene event. This temperature decrease falls within the reconstructed range in the North Atlantic region, with a larger cooling across the EOT at Sites 336, 913 and Kysing-4 (North of Site 647), and a somewhat smaller SST decrease at Site U1404 (South of Site 647) (Fig. 3).

Our SST record at Site 647 does not cover the Step 1 or EOIS events in detail (Fig. 2), but similar to the record from Site U1404 on the Newfoundland margin (Liu et al., 2018), these events do not appear to be associated with any prolonged surface temperature decrease. The surface cooling in the Labrador Sea that predates the Step 1

phase (Fig. 2) agrees with a variety of other, more coarsely resolved northern hemisphere proxy reconstructions. These data include, e.g., dust records from Central Asia (Abels et al., 2011; Sun and Windley, 2015), which indicate that the strongest cooling and continental aridification occurred between 35 and 34 Ma, respectively. Lastly, this cooling (the Late Eocene event) is detected in several deep-sea records (e.g. ODP Site 689 in the Atlantic sector of the Southern Ocean) as a transient ~0.5‰ excursion in $\delta^{18}$O and it probably coincides with a so-called 'precursor glaciation' on Antarctica (Katz et al., 2008; Hutchinson et al., 2021) interpreted to be driven by 405-kyr and ~110-kyr eccentricity minima (Fig. S3). Based on these lines of evidence, we infer that the Late Eocene event had an impact on several globally distributed locations. However, the Atlantic sector of the Southern Ocean experienced only a transient cooling of bottom and surface waters of ~1 ℃ at that time (Bohaty et al., 2012), whereas our data suggests that during the Late Eocene event surface temperatures in the vicinity of Site 647 experienced a distinctive cooling step.

## 4.2. Sea surface temperature in the North Atlantic across the EOT

The still low resolution of SST data across the EOT in the North Atlantic, compared to time equivalent benthic $\delta^{18}$O records, do not allow for any detailed analysis of the changing spatial or temporal SST patterns in the North Atlantic, or identification of sequential forcing mechanisms or leads or lags that could explain them. For example, in sites 336 and Kysing-4, where the data density is high (~35.8 Ma), the SST data has a large range in a short interval, suggesting these are highly dynamic regions and more so than site 647A (Fig. 3). In other sites like 913 there are only 6 data points between 37 Ma and 32 Ma, making it impossible to identify temporal patterns or attribute them to internal or external variability. However, we combine the available core data in an ensemble to derive an overarching picture of cooling across the 5 Myrs bracketing the EOT in the North Atlantic, (from 37 Ma to 32 Ma, Fig. 3). Specifically, we calculate the average temperature values from 37.0 Ma to 34.5 Ma ("pre-34.5" interval) and from 34.5 Ma to 32.0 Ma ("post-34.5" interval) in all existing SST records in the North Atlantic region (Supplements). The threshold of 34.5 Ma is chosen, because that is where the shift towards colder temperatures in Site 647 is recorded. We present the SST temperatures in these two intervals as a function of latitude and note that the higher latitude cores are on average colder than lower latitude cores, as one might expect (Fig. 4). The cooling across the EOT indicate polar amplification with stronger cooling in the poleward sites such as 913 and 336 (Fig. 4), although we emphasize the high uncertainty in averaging so few data points in these records.

## 5. Data-model comparison and implications

### 5.1 Absolute sea surface temperature values in the North Atlantic between 37 and 32 Ma

Here we compare the late Eocene (37 to 34.5 Ma) and early Oligocene (34.5 to 32 Ma) SST at the five North Atlantic core sites to the four combinations of an open and closed Arctic, and 400 and 800 ppm atmospheric $CO_2$ concentrations as described in Hutchinson et al. (2018, 2019). The selected time frame from 37 to 32 Ma covers the most complete data-derived SST evolution from all selected sites (Fig. 3). Most of the simulations do a reasonable job at matching proxy SSTs at lower latitudes but none of the simulations can produce the warm proxy-derived SSTs in the northern North Atlantic during the late Eocene (Fig. 4, 6), suggesting that the model has too low high latitude temperatures for the late Eocene. There may be several possible explanations for this. The applied $CO_2$ concentration of 800 ppm may still be too low for the late Eocene. The existing $P_{CO_2^{atm}}$ reconstructions across the EOT are of low resolution and are characterized by a large range of absolute values and relatively high levels of uncertainty (c.f. Anagnostou et al., 2016; Steinthorsdottir et al., 2016; Zhang et al., 2013). However, this is probably not the main reason for high latitude warmth in the records compared to the model because *i*) it is unlikely that the $P_{CO_2^{atm}}$ was much more than 1000 ppm in the late Eocene (Fig 3a), and *ii*) higher $CO_2$ concentration also implies somewhat higher low-latitude temperatures which are not underestimated in the current model simulations. Alternatively, it may be that the TEX$_{86}$-derived SST data is warmer than the model output because it represents a summer signal. Several studies of TEX$_{86}$-derived SSTs of the Eocene greenhouse state suggest the possibility of a summer bias at higher latitudes (e.g. Davies et al., 2019; Hollis et al.,

2012), and the summer SSTs are indeed a better match for the proxy data (Fig. S4). While some degree of seasonal bias cannot be ruled out, the overall trends and absolute SST estimates from the TEX$_{86}$ proxy in our record correspond well with those of $U_{37}^{K\prime}$ (Fig. S2). The $U_{37}^{K\prime}$ proxy is derived from haptophyte algae, which generally have different bloom periods than Thaumarchaeota, and are thought to reflect annual mean or spring SST (Müller et al., 1998). This argues at least against a strong seasonal bias in the $U_{37}^{K\prime}$ or TEX$_{86}$ records. Alternatively, the model has too cold high-latitude temperatures either because of too low climate sensitivity to CO$_2$ or insufficient polar amplification due to inadequate cloud feedbacks (Baatsen et al., 2020; Lunt et al., 2021). The simulation with the higher 800 ppm CO$_2$ and the closed Arctic (with active AMOC) gives the warmest absolute temperature in the North Atlantic and is therefore the closest to proxy records both for the late Eocene (37 to 34.5 Ma) and early Oligocene (34.5 to 32 Ma) intervals (red dashed line in Fig. 5, Fig. 6d).

**5.2 Sea surface temperature change across the EOT in the North Atlantic**

As detailed above, the EOT cooling is usually attributed to a decrease in atmospheric CO$_2$ (see summary in Hutchinson et al., 2021). Numerous studies have suggested that accelerated CO$_2$ decline may have been triggered by the start-up of the AMOC at or just prior to the EOT. The open and closed Arctic simulations shown here are part of one such study in which the North Atlantic deep-water formation is activated through closing the ocean gateways across the Nordic Seas transporting low salinity Arctic waters to the Atlantic Ocean. It results in salinification and densification of surface waters (Hutchinson et al., 2019). The AMOC is known to transport heat northward in the modern Atlantic and a freshwater-induced AMOC collapse in a modern climate state leads to cooler North Atlantic SSTs (Jackson et al., 2015). In our EOT simulations a start-up of the AMOC through closing the connection to the Arctic is associated with a >5 °C temperature increase in some locations of the Nordic Seas (Fig. 7a,b), suggesting a similar role for the AMOC in northward heat transport during this period. Some of the warming could also be due to reduced heat transport to the Arctic in the closed Atlantic-Arctic gateway scenario. The warming from the AMOC start-up is greater in the colder 400 ppm climate than the warmer 800 ppm climate, but this could be simply because the 400 ppm Arctic-closed simulation is further from equilibrium than the other simulations, with the AMOC still weakening and the SST at the core-site still cooling at the time of analysis. The cooling from a reduction in atmospheric CO$_2$ is of similar magnitude as the AMOC warming, albeit slightly weaker and with a different spatial pattern, reaching further south into the Subtropical gyre (Fig. 7c,d). This cooling trend is stronger when the connection to the Arctic is open and the AMOC is off, but again, this could be because the Arctic-closed 400 ppm case has not cooled to equilibrium yet.

To investigate whether greenhouse cooling could compensate for AMOC warming at the EOT, we compare the 800 ppm Arctic open simulation with the 400 ppm Arctic closed simulation (Fig. 7e). While we observe an overall cooling in the Arctic and Subtropical gyre, there is heterogeneity with the Subpolar gyre remaining warm. While this could be due to the 400 ppm Artic closed simulation still cooling at this point in the analysis, without a longer run it cannot be concluded for certain that the reduction recorded by the proxies would be matched by the model simulations. To complicate matters, at the higher northern latitudes, where temperature anomalies in both simulations and proxy reconstructions are largest, the data is also the sparsest (Fig. 4). Nevertheless, a real data-model mismatch should be considered and explanations for it explored. The first possibility is that the AMOC did not start up at or just prior to the EOT but had started earlier, e.g. in the middle Eocene (Boyle et al., 2017; Vahlenkamp et al., 2018) and intensified 500 kyr prior to the EOT (Coxall et al., 2018). The change in heat transport from AMOC strengthening should be weaker than from a complete cold start up. Alternatively, the changes in Arctic gateways bathymetry could have been subtler in reality than in the model. This would dampen the impact on the circulation and SST, or the AMOC may have started up through an altogether different mechanism such as the widening of the Southern Ocean Gateways (Elsworth et al., 2017) which could have a smaller warming effect. This latter process of starting up the AMOC did not work in the modelling study of Hutchinson et al. (2019), but such results can be model dependent and requires corroboration. There are also model deficiencies that could explain the overall North Atlantic warming, such as the above mentioned exaggerated meridional temperature gradient in the model at the EOT (causing too much heat transport through the AMOC) and too low climate sensitivity to the CO$_2$ decrease. Another point to mention is that the model produces deep water in the Labrador Sea and the Greenland Sea when the Arctic is closed-off (Hutchinson et al., 2019). Yet there is no evidence in Site 647 records for deep water formation in the Labrador Sea before or directly after the EOT (Cramwinckel et al., 2020; Coxall et al., 2018). The model AMOC is therefore feeding

deep water from two regions instead of one, and thus could be too strong. Another possibility is that the $CO_2$ decline at the EOT was greater than suggested by existing proxy records (e.g. Anagnostou et al., 2016; Zhang et al., 2013), in which case $CO_2$-related climatic cooling at northern high latitudes at the EOT could have been more extreme than currently assumed. However, while it is reasonable to assume that the pre-EOT $CO_2$ was higher than 800 ppm, there is little evidence that it may have been as low as 400 ppm after the EOT (Fig. 3a). These
explanations remain speculative and requires further investigation in a modelling-focused study.

### 5.3 Sea surface temperature variability across the EOT at site 647

With its higher temporal resolution compared to other North Atlantic records at the time, it is interesting to note some temporal signals in the SST at Site 647 across the late Eocene. In particular, our data suggest that there may be a temperature minimum at ~35.7 Ma and a maximum at ~34.9 Ma, followed by the cooling step (Fig 3).
The SST variability described at Site 647 is well resolved, even though the minimum and the maximum are based on one or few data points. The late Eocene SST at Site 647 seems reasonably stable, considering that data points that are close together in time have similar SSTs (Fig 3). The increase in SST between 35.7 Ma and 34.9 Ma could possibly be due to an increase in the AMOC, culminating at 34.9 Ma (Coxall et al., 2018). Thereafter, normal background $CO_2$ cooling could have resumed. A peak in SST was also present during this time at low
latitude Atlantic Site 959 (Cramwinckel et al., 2018) and in the North Sea (Śliwińska et al., 2019). Other Atlantic SST records are of insufficient resolution to study this type of variability, so this point remains speculative. Higher resolution SST records from the east and west northern North Atlantic and Nordic Seas spanning the late Eocene would be desirable to fully address this hypothesis.

Today Site 647 is located in the south-western part of the North Atlantic Subpolar gyre, influenced by cold and
low-salinity subarctic surface waters. The barotropic streamfunction in the model, a combination of the wind-driven gyre transport and the meridional overturning streamfunction, suggests that at the EOT the site was in or near the boundary of the Subtropical gyre and the Subpolar gyre and that this region was highly dynamic (Fig. 8). The horizonal circulation in this region changes dramatically when the Arctic closes and the AMOC starts up, with the streamfunction-derived Subtropical gyre reaching more northward into the Labrador Sea and the
Subpolar gyre moving closer to the Western Boundary (Figs. 1, 8). The result is a switch of the mean current direction at the site location from north-eastward to south-eastward. The North Atlantic warming associated with the closing of the Arctic, broadly outlines the Subpolar gyre boundary of the open cases and it has a strong gradient at Site 647 so that if the site was just a few degrees to the south (or arguably the gyre to the north) it would experience much less warming and might even have a degree or two cooling when taking into account the
expected $CO_2$ cooling at the EOT (Fig 7a,b,). The position and strength of the gyres, as well as the strength of the AMOC, is likely model dependent, and should not be taken too literally. However, suffice to say that it depends critically on the paleogeography of the region which was dynamic at the time (Hutchinson et al., 2019). Even a globally homogenous forcing factor such as $CO_2$ results, through regional feedbacks, in heterogenous changes in the North Atlantic SST (Fig 7).


### 6. Conclusions

Our new SST record derived from organic geochemical paleo-thermometers provide the highest resolution of SST across the EOT in the northern North Atlantic to date. Our SST record show variability in the 2.5 Myrs leading up to the EOT, which includes a ~ 800 kyr warming interval before the final cooling step which took
place ~500 kyr before the EOT. Model simulations of various possible paleogeographic and atmospheric $CO_2$ scenarios at the time indicate that the site is located in a dynamic region close to the Subtropical and Subpolar gyre boundary. Atmospheric $CO_2$ or paleogeographic changes would change the gyre location, strength, and structure. It could even change the direction of the mean current at the site, influencing the local SST. Whatever the driver, our model suggests that there is usually some coherence in the North Atlantic SST response across the
Subpolar gyre and separately the Subtropical gyre, but in general the response is heterogeneous across the North Atlantic. Any extrapolation of ocean warming or cooling at a specific site location to the wider Atlantic and global climate drivers should therefore be done with care.

In order to compare the SST changes across the EOT with other North Atlantic lower resolution records, the SST was averaged over a late Eocene bin spanning the 2.5 Myrs before the 34.5 Ma cooling step in Site 647 (37 – 34.5 Ma), and the early Oligocene bin spanning the 2.5 Myrs after this step (34.5 – 32 Ma). In this basin wide view, the cooling at the EOT is found to be larger at higher latitudes, although this is also where data is particularly sparse. The binned data was compared to four model simulations of EOT scenarios with high (800 ppm) and low (400 ppm) atmospheric $CO_2$, and open and closed Arctic-Atlantic gateways, also representing off and on AMOC scenarios, respectively. The cooling across the EOT is best simulated with a drop in $CO_2$ alone. However, several deep ocean circulation proxies suggest that the AMOC started up just prior to the EOT (Coxall et al., 2018), and our model simulations indicate that if the AMOC starts up (through Arctic closure in our case), the $CO_2$ cooling is approximately countered by warming from the increased heat transport. However, several caveats need to be raised when making such a comparison. Our AMOC-on 400 ppm simulation is still cooling and it is not possible to know the final SST state. But suffice to say, the final state would be cooler than the one shown here, especially in the high latitude regions that is sensitive to the AMOC which is still decreasing rapidly at the time of the analysis. It is possible that the AMOC did not start up in the late Eocene, but alternative explanations are then required for the deep ocean proxies that suggest this (Coxall et al., 2018; Hutchinson et al., 2021). Also, if the EOT cooling was driven by stand-alone $CO_2$ changes, the question remains, why was there a sharp deep ocean cooling step before the Antarctic ice-sheet growth (Lear et al., 2008). Other possibilities are (*i*) that the AMOC started up earlier and just intensified at the EOT, (*ii*) that the model is overestimating the AMOC heat transport due to too warm high latitude temperature at the Eocene, (*iii*) that the $CO_2$ decrease is larger than modelled here, or (*iv*) that the model has too low sensitivity to $CO_2$ cooling. It should be noted that the model time slices present only a few possible scenarios of which none were probably an exact reality at any point. The pre- and post-EOT world would not correspond to any single scenario but would be a dynamic time of variable paleogeography and $CO_2$. It is also worth emphasizing again the model-dependence of these results, as the ocean circulation and stratification varies greatly between models of the Eocene, even when forced with a similar set of boundary conditions (Zhang et al., 2022).

Our new data aids in understanding of the timing and the spatial pattern of temperature changes related to the transition into the unipolar icehouse climate state. The model simulations highlight the heterogeneity of North Atlantic SST and its response to different forcing factors. This calls for more proxy records to increase the spatial coverage and resolution of regional temperature trends across the North Atlantic in order to identify possible thermohaline fingerprints of the AMOC start-up at the EOT. For areas located south from site 647 and Kysing-4 this could include construction of east-west Atlantic surface and deep water $\delta_{18}O$ and temperature gradients using multiple palaeotemperature proxy methods (e.g. clumped isotopes, foraminiferal Mg/Ca, or $TEX_{86}$). For the higher northern latitudes, where calcareous microfossils fossils are very limited in this time interval, this could include higher resolution SST proxy reconstruction based on $TEX_{86}$ and/or $U_{37K'}$. Despite the existing hiatuses at the EOB interval in the North Atlantic region, increasing sampling resolution in the existing sites in the interval from 37 to 32 Ma would be beneficial. . A formal model intercomparison project (e.g., EOT-MIP) to compare the response in a variety of different EOT models would increase our confidence of the ocean and climate system's response to proposed drivers of the EOT, and thus facilitate for more robust model-data intercomparisons.

**Acknowledgments**

This research was funded by the Danish Council for Independent Research/Natural Sciences (DFF/FNU; grant 11-107497) to K.K.Ś., Swedish Research Council (VR) grants awarded to A.M.dB (2016-03912 and 2020-04791) and H.K.C. (2008-2859), Formas grant to D.K.H. (2018-01621) and the Netherlands Earth System Science Centre (NESSC) and the Ministry of Education, Culture and Science (OCW) to S.S.. The model simulations were enabled by resources provided by the Swedish National Infrastructure for Computing (SNIC) at the National Supercomputer Centre (NSC), partially funded by the Swedish Research Council through grant agreement no. 2016-07213. We thank Walter Hale at Bremen Core Repository (BCR) for collecting samples. We appreciate inspiring discussions with J. Firth and J. Backman and laboratory assistance from A. Metz. This research used samples provided by the Ocean Drilling Project (ODP). ODP was sponsored by the U.S. National Science Foundation and participating countries under management of Joint Oceanographic Institutions.

**Availability of data**

The supplementary information is available in the Supplementary Information file and raw data are hosted at XXXX. The model data used in this analysis will be made available upon publication in an open access database hosted by the Bolin Centre for Climate Research (https://doi.org/10.17043/hutchinson-2022-eocene-oligocene-1)

**Authorship contribution statement**

K.K.Ś. designed the research. K.K.Ś and S.S. generated organic geochemical proxy ($TEX_{86}$, $U^{K'}_{37}$) data. H.K.C helped to produce the Site 647 age model and correlate with IODP Site 1218. D.K.H. ran all model simulations. K.K.Ś and A.M.dB were the main authors of the manuscript, although all authors contributed with data interpretation and writing.

**Author information**

The authors declare no competing financial interests. Correspondence should be addressed to K.K.Ś. (kksl@geus.dk)

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

**Figure captions**

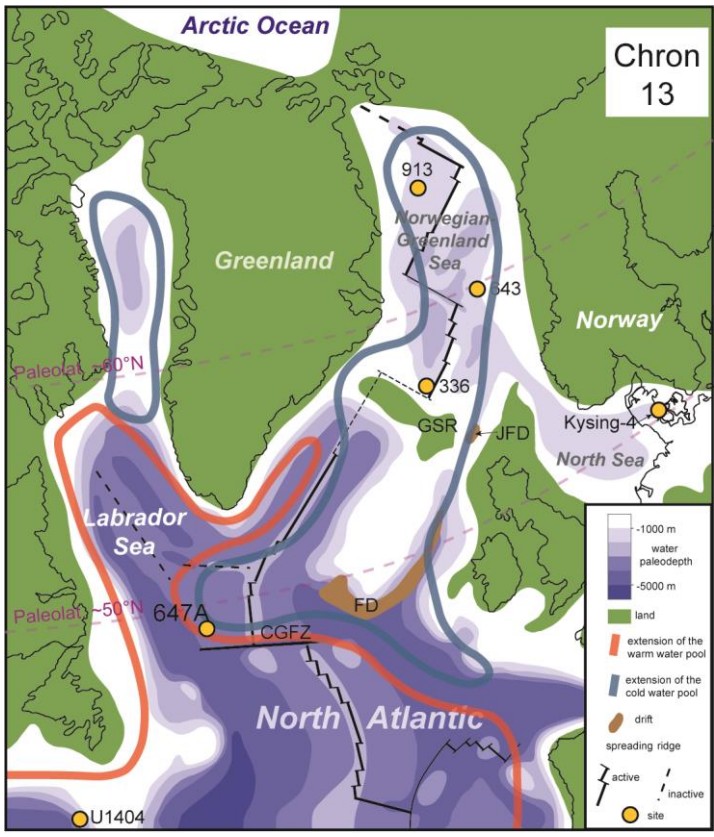

**Figure 1:** The late Eocene (magnetic polarity Chron 13; 33.705–33.157 Ma (GTS2012)) location of Site 647 (ODP Leg 105) and other sites studied for temperature proxies (pollen in ODP 913B, ODP 643, ODP 985 (Eldrett et al., 2009); alkenones in DSDP 336, ODP 913B and IODP U1404 (Liu et al., 2009, 2018); GDGTs in Kysing-4 (Śliwińska et al., 2019)) referred to in the text. The paleogeographic map is modified after Arthur et al., (1989), Piepjohn et al., (2016), Śliwińska et al., (2019) and references thein. Abbreviated oceanic features identified are: the Feni Drift (FD)(Davies et al., 2001), the Judd Falls Drift (JFD) (Hohbein et al., 2012), Greenland–Scotland Ridge (GSR), and Charlie-Gibbs Fracture Zone (CGFZ). The red and blue lines labelled as 'extensions of the warm water pool' and 'extension of the cold water pool', respectively, represent the positions of surface ocean gyre systems that expand with a late Eocene AMOC switched on in our model experiments.





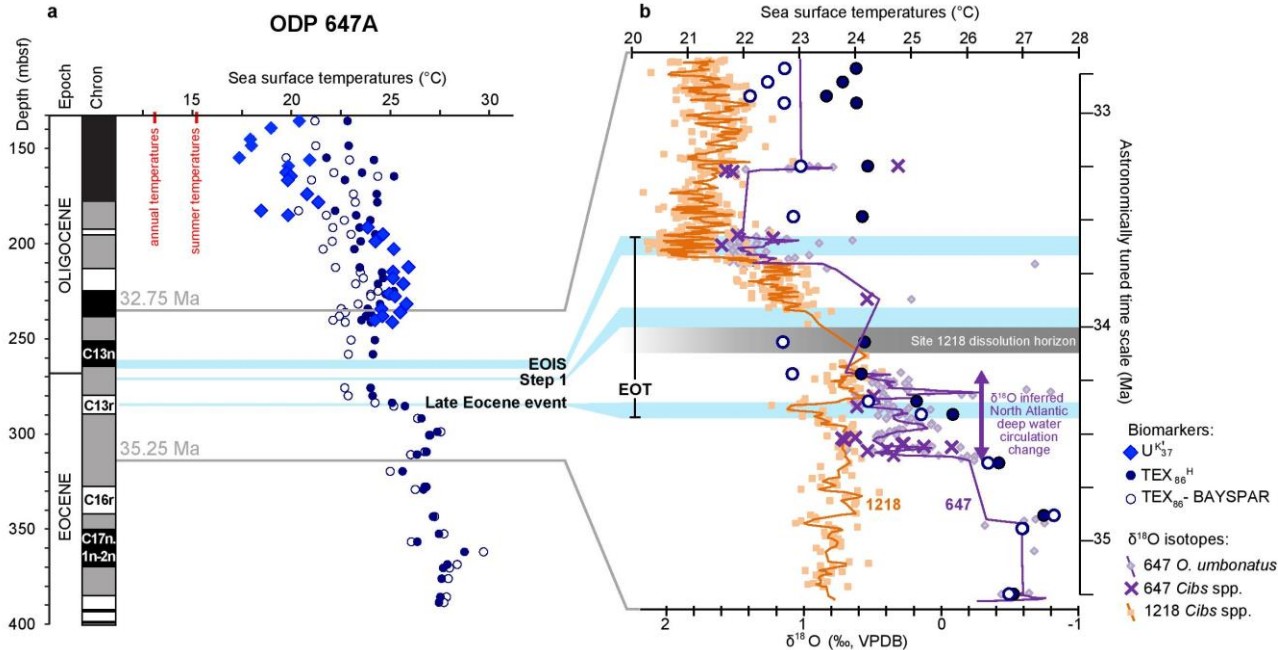

**Figure 2:** The sea surface temperature (SST) record from the Ocean Drilling Program (ODP) Site 647A. a) SSTs based on $TEX_{86}$ and $U^{K'}_{37}$ indices (this study). Magnetostratigraphy after Firth et al., (2013). MAT – modern average annual temperatures (10.6 °C), ST – modern summer temperatures (15.2 °C) at the paleolocation of 46°N based on the Ocean World database. EOIS – Earliest Oligocene Isotope Step. EOIS, Step 1, Late Eocene event and EOT follows nomenclature after Hutchinson et al. (2021) b) The new temperature record across the Eocene-Oligocene transition (EOT) compared to (*i*) benthic foraminifera oxygen stable isotope ($\delta^{18}O$) records from ODP Site 647 (*Oridorsalis umbonatus*; > 63 µm) (Coxall et al., 2018) and an inferred zone of acute North Atlantic deep water circulation change, and (*ii*) benthic $\delta^{18}O$ record from ODP Site 1218 providing the chemostratigraphic framework that allows us to extrapolate the EOIS, Step 1 and Late Eocene events to Site 647 (Coxall and Wilson, 2011). All ages are based on the GTS2012 (Vandenberghe et al., 2012).

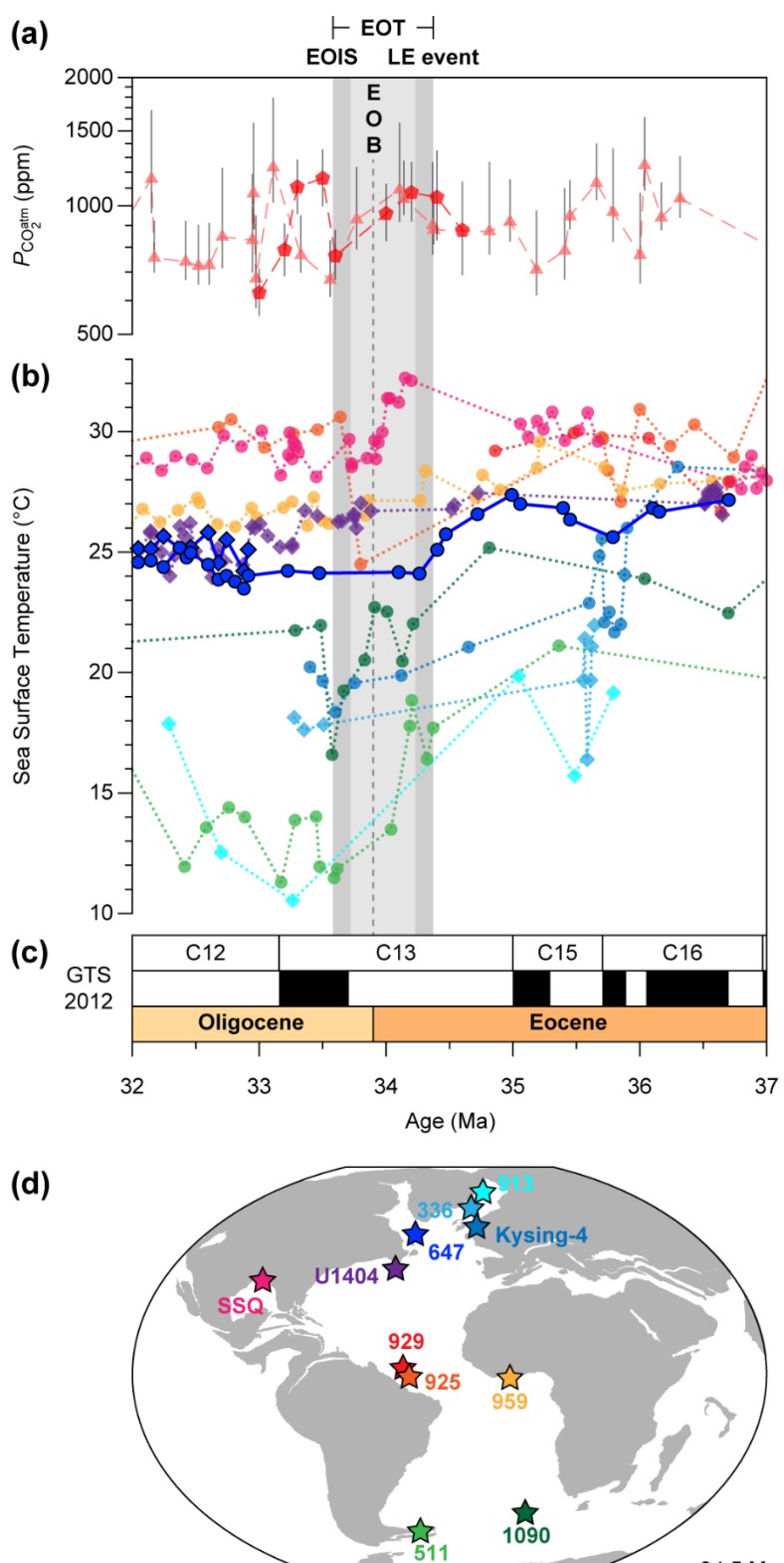

**Figure 3:** SST evolution across the EOT in the Atlantic Ocean. a) Reconstructed $P_{CO_2^{atm}}$ based on planktonic foraminiferal $\delta^{11}B$ (pentagons) (Pearson et al., 2009) and phytoplankton alkenone $\delta^{13}C$ (triangles) (Pagani et al., 2011; Zhang et al., 2013). The effect of $P_{CO_2^{atm}}$ on radiative forcing scales logarithmically. b) Newly generated and published (Cramwinckel et al., 2018; Houben et al., 2019; Inglis et al., 2015; Liu et al., 2018, 2009; Śliwińska et al., 2019; Wade et al., 2012) reconstructed SSTs based on $U_{37}^{k\prime}$ (diamonds) and $TEX_{86}^{H}$ (circles). All ages are converted into the GTS2012 (Vandenberghe et al., 2012). c) Magneto- and chronostratigraphy based on the GTS2012 (Vandenberghe et al., 2012). d) Paleogeography at 34.5 Ma (https://www.odsn.de/) with color-coded site locations of the SST records shown in panel B. SSQ stands for St. Stephen's Quarry.

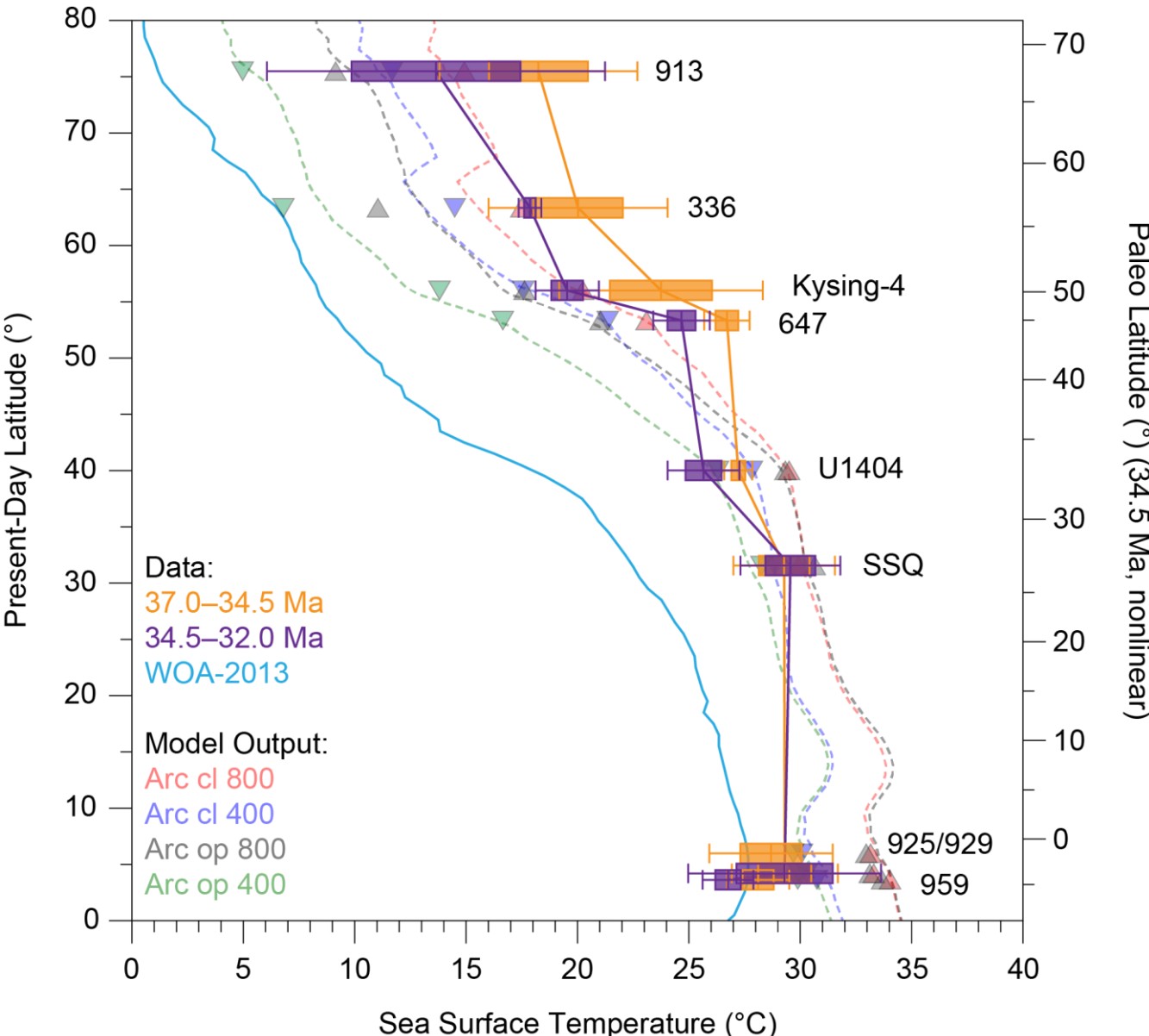

**Figure 4:** Data-model comparison of latitudinal SST gradients for the late Eocene (37 – 34.5 Ma, orange bars and solid lines) and early Oligocene (34.5 – 32 Ma, violet bars and solid lines) states to the four different model simulations (raw data are shown on Fig. S5).. The dashed lines show the zonal average SSTs at that latitude in the Atlantic sector and the triangles show the site specific temperatures in the simulations. The 1 and 2 sigma error bars are indicated around the data points. On the left of the figure in solid blue line is the zonally average present-day Atlantic SST from the World Ocean Atlas (Boyer et al., 2013), used as a reference for the present-day Atlantic sea surface temperature latitudinal gradient. Model paleolatitudes (right hand axis) are shifted with respect to present-day latitudes of the data (site and WOA data, left hand axis) by the average offset of –7.0° (error: ± 1.5°) for the sites considered. For sites 913, 336 and U1404 SST data in derived from $U^{k\prime}_{37}$, while in sites Kysing-4 and 647 SST is derived from $TEX^{H}_{86}$. Arc op - Arctic-Atlantic gateway open; Arc cl - Arctic-Atlantic gateway closed; 800 – 800 ppm $CO_2$ simulation; 400 – 400 ppm $CO_2$ simulation.

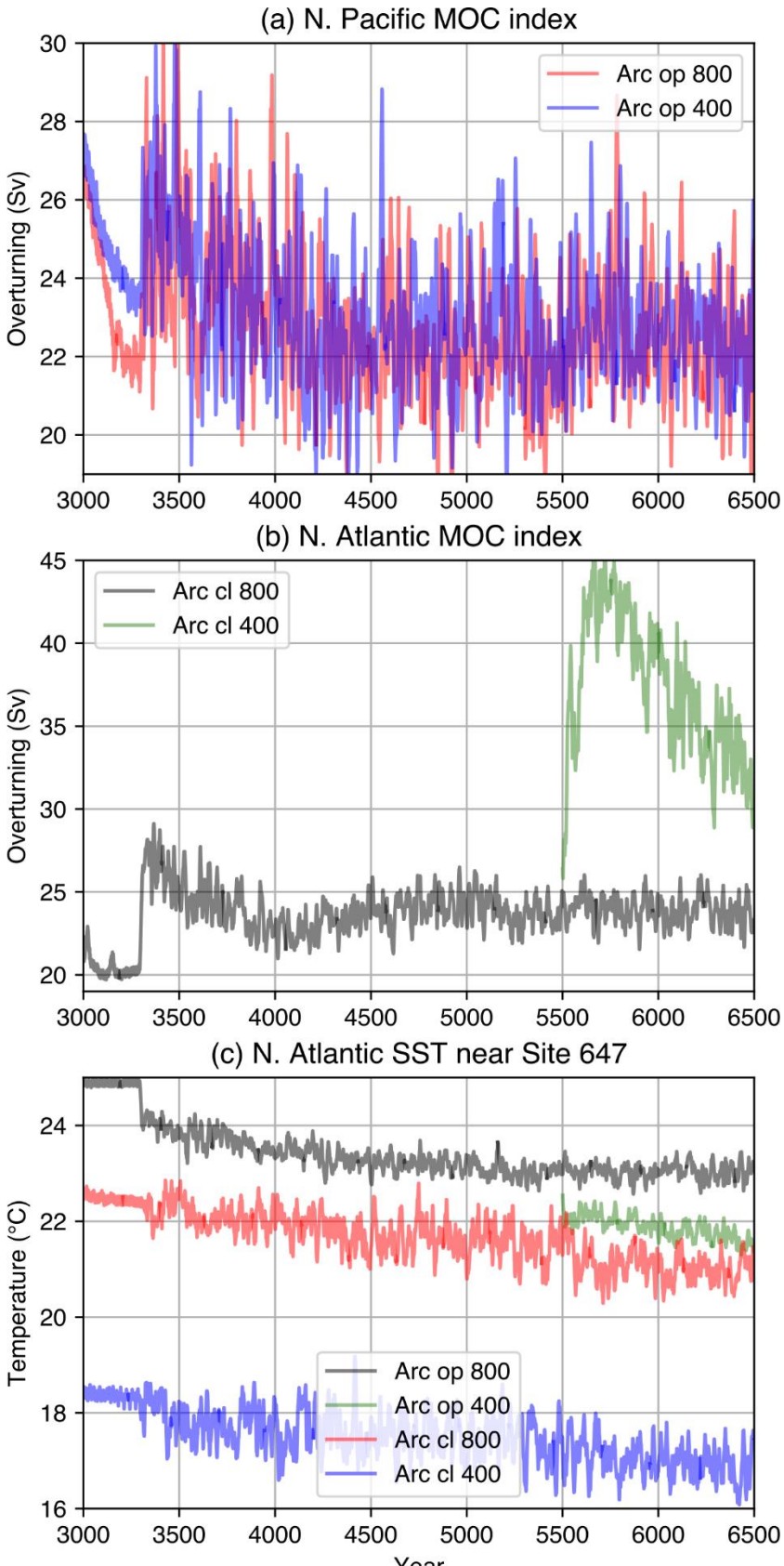

**Figure 5:** Time series of the North Pacific meridional overturning circulation (MOC) index (a), the North Atlantic MOC index (b), and the SST averaged over a 5°x5° box around core site 647 in the four model simulations.

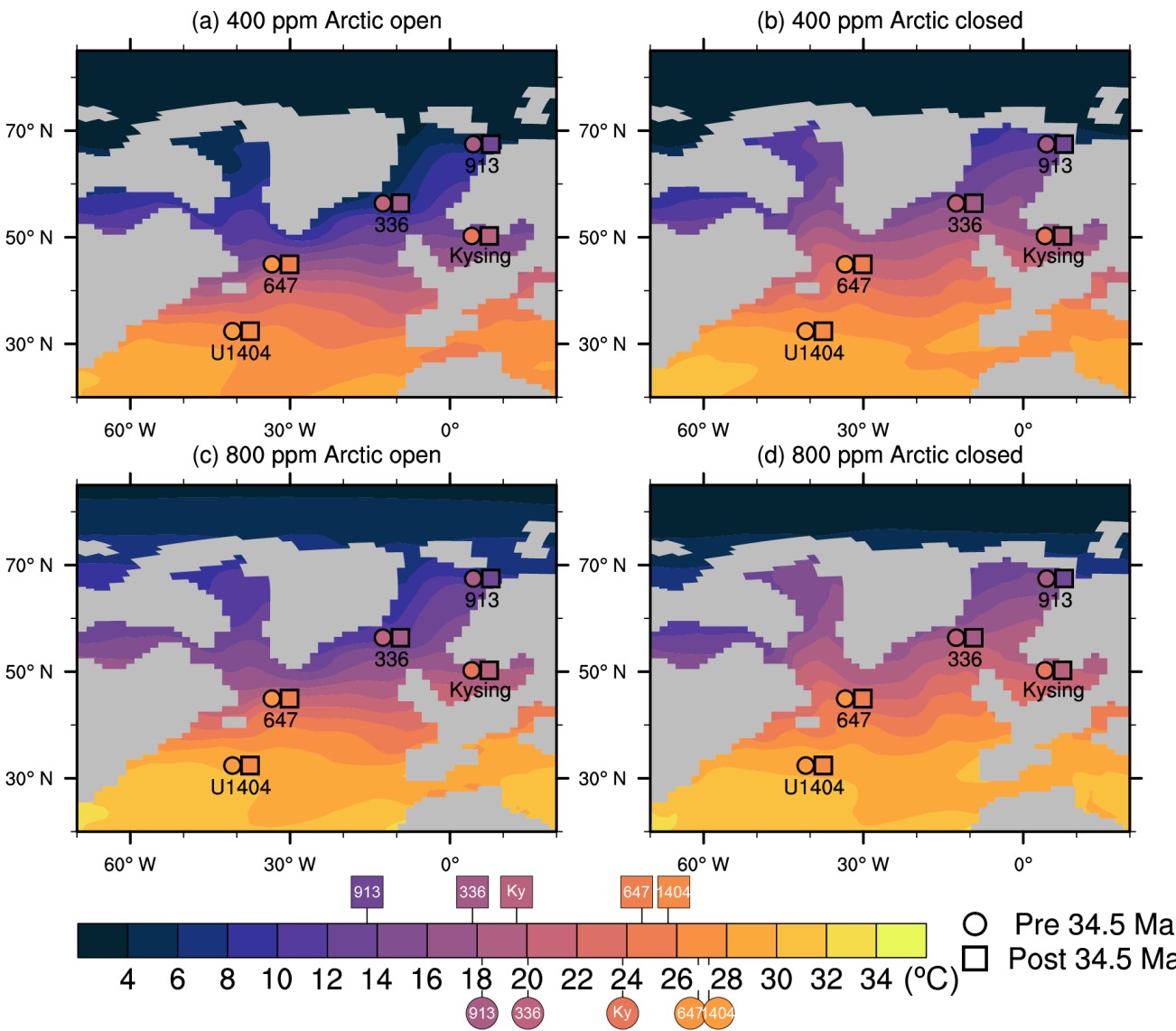

**Figure 6:** Comparison of model temperatures in the four simulations in the North Atlantic with late Eocene (circles; 37 – 34.5 Ma) and early Oligocene (squares; 34.5 – 32 Ma) proxy data (SST derived from $U_{37}^{k\prime}$ in sites 913, 336 and U1404, and SST derived from $TEX_{86}^{H}$ in sites Kysing-4 and 647). Contours show the modelled annual mean SST for the Arctic open (a, c) and the Arctic closed run (b, d) for atmospheric $CO_2$ concentrations of 400 ppm (a, b) and 800 ppm (c, d). The coloured circles show the proxy data averaged between 34.5 and 37 Ma (late Eocene) and the coloured squares show the proxy data averaged between 34.5 and 32 Ma (early Oligocene). For all sites: the average SST proxy records and the modelled SST for each of the four climate scenarios are shown also in Figs. 4, S5 and in the Supplementary Information file.

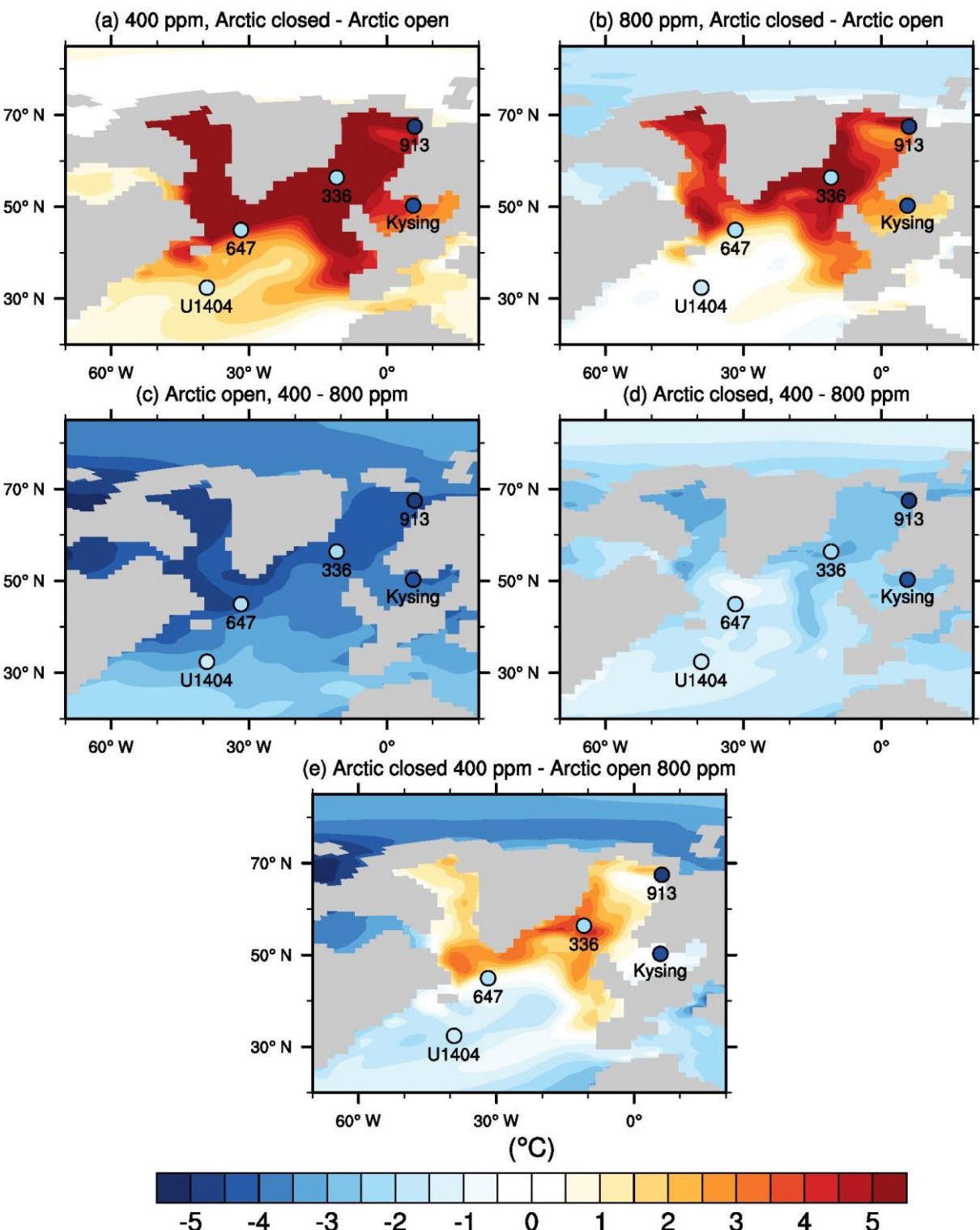

**Figure 7:** Site specific SST anomalies across the EOT from proxy data (SST derived from $U^{k\prime}_{37}$ in sites 913, 336 and U1404, and SST derived from $TEX^{H}_{86}$ in sites Kysing-4 and 647) compared with SST differences between the model simulations. Shown is the SST impact of closing of the Arctic for a 400 ppm climate (a) and an 800 ppm climate (b) as well as the impact of reducing $CO_2$ from 800 ppm to 400 ppm when the Arctic is open (c) and when it is closed (d). The final subplot shows the difference between the 800 ppm open Arctic and the 400 ppm closed Arctic (e). The coloured circles show the SST change ($\Delta$SST) for each site across the EOT as suggested by the proxy data records. $\Delta$SST is calculated as the difference between the pre 34.5 Ma (late Eocene) SST average, and the post 34.5 Ma (early Oligocene) SST average (Fig. S5 and Supplementary Information File).

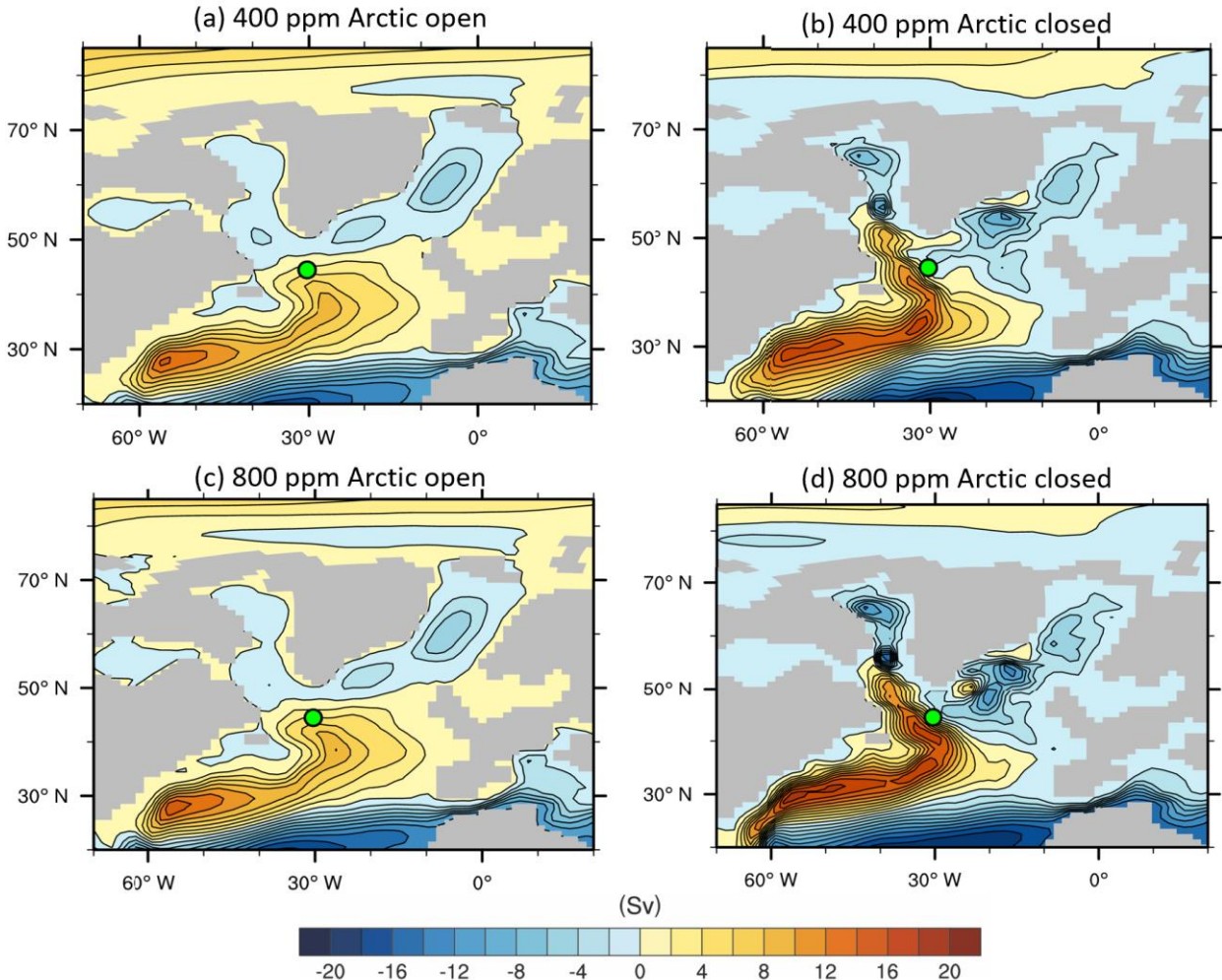

**Figure 8:** Barotropic streamfunctions illustrating the horizontal circulation (positive = clockwise) for the Arctic open (a, c) and the Arctic closed run (b, d) for atmospheric $CO_2$ concentrations of 400 ppm (a,b) and 800 ppm (c,d). The contour interval is 2 Sv. Note the modified and intensified subtropical and subpolar gyre systems in the Arctic closed experiment and the critical position of Site 647 at their boundaries.