# Peer review of "Sea surface temperature evolution of the North Atlantic Ocean"

_Climate of the Past, 2021_

## Author Response (AR1)

Clim. Past Discuss., author comment AC1
https://doi.org/10.5194/cp-2021-184-AC1, 2022

[Figure]

**Reply on RC1**

Kasia K. Śliwińska et al.

Author comment on "Sea surface temperature evolution of the North Atlantic Ocean across the Eocene-Oligocene Transition" by Kasia K. Śliwińska et al., Clim. Past Discuss., https://doi.org/10.5194/cp-2021-184-AC1, 2022

**We would like to thank Dr. Michiel Baatsen for his time and effort reviewing our manuscript. Our replies to the comments are marked as bold text.**

The authors present a new high quality dataset of proxy SSTs in the Northwest Atlantic Ocean spanning the late Eocene into the Oligocene. The obtained SSTs are compared to other available sites nearby, as well as adequate recent model results which spark an interesting discussion. I believe that these results can greatly contribute to our understanding of the events surrounding the EOT, covering a region where the available data is scarce.

**Thank you for such a positive overall evaluation of our manuscript**

General remarks:

The manuscript is structurally sound, provides a balanced amount of proxy-/model-derived data as well as an extensive methodology and discussion sections. Overall, an extensive language/grammar check is needed as many small errors and typos remain in the text. **We will go through the manuscript and corrected the language and grammar errors and typos.**

While the comparison to model data is quite useful and important, the use of a single model makes it hard to judge which part of the results are the most model-sensitive. It is probably quite hard to find other adequate model data for this specific case, but the limitations could be highlighted more, especially considering the different deep water formation regimes. Compared to e.g. results in DeepMIP, despite representing Early Eocene conditions, it is clear that different models present a whole suite of possible overturning regimes under comparable boundary conditions. **This is true. We will now cite Zhang et al. (2022) and we will discuss how the GFDL model overturning circulation in the Eocene compare to other models and what the implication of that could be for the results.**

Although relevant, much of the discussion is rather superficial and qualitative in nature. Many of the claims/ideas would be hard to check in the available model data, but it could be helpful to have a more detailed look into some of the mechanisms that drive the

changes in circulation and SST shown. This could include e.g. the radiative balance, surface/gateway fluxes, meridional temperature gradients and/or transports, and wind stresses. **The reason why we have not presented extensive model analysis is because we do not want to take the focus away from the core contribution of the paper, which is the new high temporal resolution SST data in a region near modern deep-water formation zones. We do not mean to make claims; but on the contrary, present how difficult it can be to use paleoclimate proxies and modelling to make any strong claims about what happened at the EOT. However, we can see that at some points we used words like "probably because of" without justifying that. We will be more careful with that and add analysis such, as further analysis of the AMOC stream function, where appropriate. Please also see below how we address the specific points relevant to this general point.**

Minor comments:

The introduction cites many earlier, i.e. pre-2010 papers, so it could be useful to check for some more recent work on some of the subjects discussed here (e.g. potential Greenland glaciation or sensitivity to model boundary conditions around the EOT). **We will check it.**

In equation 1, the terms [37:2] and [37:3] are not explained **We will add an explanation of these terms in Equation 1**

L184: rephrasing this sentence would make it more readable **it will be corrected**

L194: this statement may be a bit outdated, as several studies presented model simulations of comparable resolution in more recent years; e.g. Li et al. 2018, Tardif et al. 2020, Baatsen et al. 2020. Also several studies within DeepMIP, albeit for Early Eocene. **We will now state that our model resolution is in line with state-of-the-art models for the EOT and add these citations.**

L209: I would rephrain from using the term 'observations' for proxy data; these are proxies that give us an indication of SSTs in the deep past, but are not actual observations. **It will be corrected, we will refer to them as "proxy reconstructions"**

Section 4.2: same remark as L209, also acronym NA used for North Atlantic which seems inconsistent with the previous sections, in which no acronym is used or introduced. **The acronym is introduced in line 27. We will now however, spell North Atlantic in full**

L266: increased $CO_2$ would indeed likely lead to higher temperatures and a reduced meridional gradient, but will still increase lower latitude temperatures as well. It does seem like increasing $CO_2$ would not be very helpful if only higher latitude SSTs are underestimated by the model. **We agree that the model already overestimates the low latitude temperatures by a few degrees when the CO2 is 800 and this will get worse if the CO2 is increased. We have in the previous version pointed to the fact that this higher CO2 explanation has a problem in that there is little evidence for a much higher CO2 and now we will add the issue about increasing the lower latitude temperature as well.**

L286: As the AMOC has not collapsed in the modern climate, this comparison is a bit odd. Some future projections show an AMOC slowdown, but this needs to be stated more carefully. **The reference is to an idealised study in which the AMOC is artificially shut down by a salinity perturbation. We do not mean to suggest that an AMOC collapse is a modern observation, only that in a modern framework, an induced AMOC collapse also cause NA cooling. We will clarify this in the new revision.**

L289: it would be helpful to have some more specifics on the actual related AMOC strengths and associated meridional heat transport to support this claim. **We will add further analyses of AMOC streamfunctions, and the SST in the formation regions in each case, in order to clarify this statement about changing AMOC strength.**

L327: This may even be more important than changes in AMOC strength, as they do not induce the expected SST changes seen in the proxy record (as argued). On the other hand, the idea of this site sitting on the edge of 2 gyres shifting over time does not match well with the relatively small SST variability, as such SST changes are often strongly related to the background gradients. **Indeed, the zonal temperature gradients are in general weak in the model, so a simple shift in the gyre boundaries would not make a big SST difference at the boundary of the gyres when all else is the same. The SST gradient can be larger between the western boundary current and the eastern side of the same gyre than between two gyres. We will now clarify this aspect in more detail. Specifically, the barotropic streamfunction in Figure 7 shows that the circulation in this region changes completely when the Arctic closes and the AMOC starts up, and Figure 6a and 6b show that the associated NA warming broadly outlines the subpolar gyre boundary of the open cases and it has a strong gradient at site 647 so that if the site was just a few degrees to the south (or arguably the gyre to the north) it would experience much less warming and might even have a degree or two cooling when taking into account the $CO_2$ cooling. Given that the gyres are most likely quite model dependent, we do not want to put too much weight on the exact position of the site in relation to the gyres but rather point to the fact that the NA warming is quite regional and the circulation at the core site here is quite dynamic and on the boundary of where warming occurs. We will be clearer about that in the revised version of our manuscript.**

Figures:

Fig4: Not very intuitive and tough to read; many overlapping points and lines. Consider separating the pre/post-34.5Ma into 2 panels? **It is not obvious how to separate the model simulations in pre/port-34.5 Ma bins, especially the Arctic-close high CO2 case and the Arctic-open low CO2 case. We have instead significantly increased the resolution of the figure which will be much sharper in the revision.**

Fig5: It is hard to distinguish between the different panels and see the differences, especially for the proxy SSTs. Differences are also shown in Fig6, but as of now Fig5 does not seem to be adding much information apart from an overall idea of absolute SSTs. **The idea is to spatially compare the different model simulations to the proxy reconstructions for absolute SST for both pre 34.5 Ma and post 34.5 Ma, something which is not possible to see in our other figures.**

Fig7: What is the added value of showing the barotropic stream function? This is rather hard to interpret, as it represents the depth-integrated flow, while the rest of the manuscript mostly discusses (near-)surface conditions. With the presence of a meridional overturning circulation, such as the AMOC in this case, it becomes quite hard to distinguish the wind-driven gyres within these fields. **The barotropic streamfunction is a quite common way to show the boundaries of the gyres in the North Atlantic. It gives a good indication of the upper layer circulation, including the AMOC, except arguably over strong deep boundary currents. The main point here about the upper layer circulation in the region of the site being quite dynamic and the currents changing drastically with the changing paleogeography, is unlikely to break down if the deep circulation was subtracted from the streamfunction. But it is necessary to include the full depth circulation to construct a streamfunction, because it must be non-divergent.**

Technical comments/typos:

As noted above, a thorough language check is needed. **It is checked and will be corrected**

Some examples: L95 conclude with summary, L128 as follow, L151 fiveteen, L282 studies has, L304 gateway changes?, L307 require L338 provides **we will address these issues.**

Typos:

L90: circulation **will be corrected**

L206: the the **will be corrected**

L245: equivalent **will be corrected**

L251: ensemble **will be corrected**

Section 5.1: title SST (2x) -> EOT? **will be corrected**

L293: Arctic **will be corrected**

L294: it may **will be corrected**

Fig2: therein **will be corrected**

Other small Errors:

L276: the model has too low polar temperatures? **will be corrected**

L306: Southern Ocean Gateways? **will be corrected**

L357: missing word **The sentence should state: "It is possible that the AMOC did not start up in the late Eocene, but alternative explanations are then required for the deep ocean circulation proxies that suggest this (Coxall et al., 2018)"**

[Figure]

Clim. Past Discuss., author comment AC3
https://doi.org/10.5194/cp-2021-184-AC3, 2022

[Figure]

**Reply on RC2**

Kasia K. Śliwińska et al.

Author comment on "Sea surface temperature evolution of the North Atlantic Ocean across the Eocene-Oligocene Transition" by Kasia K. Śliwińska et al., Clim. Past Discuss., https://doi.org/10.5194/cp-2021-184-AC3, 2022

**We would like to thank reviewer 2 for her/his time and effort reviewing our manuscript. Our replies to the comments are marked as bold text.**

Summary

In this study, the authors generated high-resolution sea surface temperature (SST) records spanning the Eocene Oligocene Transition (EOT) using two independent organic proxies, namely algal lipid-based UK'37 and archaeal lipid-based TEX86. The study site ODP 647 is located in the North Atlantic (NA), and is thus far the most northerly NA location with a known EOT sequence. The authors interpret both UK'37 and TEX86 as reflecting SST. Comparing their records with other published NA records show that the temperature trend across the EOT is spatially heterogenous. The authors further calculated average temperature values for 37.0–34.5 Ma and 34.5–32.0 Ma for comparison with previously published climate model output from Hutchinson et al. (2018). Proxy-derived latitudinal gradient is substantially flatter than that derived from the model output. Comparing site-specific SST anomalies further highlight data-model discrepancy; the climate model output indicates warming in the subpolar gyre in stark contrast to the cooling suggested by proxy data. The authors discuss possible factors leading to this data-model mismatch by considering uncertainties in numerous aspects of data and model output.

General comments

The topic investigated fits the remit of the journal. There is a dearth of high-resolution data across the EOT, thus the data presented by this study will make an important and timely contribution to the community. The paper is generally well-written and accessible. I did spot numerous typos in the discussion though, a thorough proof-reading before resubmission would be appreciated. Some arguments are unclear and could be further strengthened. In the following I list my major concerns which I invite the authors to consider and clarify when revising their manuscript. **Thank you for the overall positive assessment. The spelling errors will be corrected, and the concerns are addressed below.**

(1) Does the age model support the interpretation of narrow time windows like Step 1 and

EOIS?

The age model is based in part on the visual correlation between the low-res benthic d18O record from ODP647 and the high-res d18O record from ODP1218. Due to the sparse temporal coverage of the 647 d18O record, it is unclear to the reader how the authors correlated the two records. It might be helpful to mark the age tie-points in Figure 2 or add a table listing the tie-points so that the reader can judge how robust the age model is. It would also be great to add some discussion on whether the uncertainty of the age model allows any interpretation of events across the EOT.

**We thank the reviewer for their attention to the age model. The stratigraphic model across the EOT in the ODP 647 is the same as presented by Coxall et al. 2018.**

**Below, we include some explanation of the age modelling approach for clarification. We then made some text revisions that we hope both clarify age model issues in the ms text and also address the reviewer's questions about uncertainties, especially in relation to identifying the narrow events EOT-1 and EOIS.**

**There are few biostratigraphic datums at Site 647, and age constraints rely mostly on P-mag tie points (see Firth et al., 2013). The discontinuous coring at the site also creates relatively large sampling gaps (and thus depth uncertainties on P-mag reversal tie points). This is most extreme close to the E/O boundary with Core 29R having the greatest sediment disturbance eliminating any coherent P-mag signal. Thus there was a depth uncertainty of max +/- 9 m on the position of the Top C13r/Base C13n. Despite this, an age model has been produced (Firth et al., 2013). Coxall et al., 2018 adjusted this age model with reference to their benthic foraminifera $\delta^{18}O$ stratigraphy. The $\delta^{18}O$ data help 'test' and slightly refine the age model as follows: (from Coxall et al., 2018 SI section)**

**----**

**"In other EOT deep sea sequences, combined $\delta^{18}O$ and magnetic reversal stratigraphy has shown that high $\delta^{18}O$ values diagnostic of the EOGM $\delta18O$ increase ('Step-2' of Coxall et al., 2005; Oi-1 of Coxall and Wilson, 2011, Zachos et al., 1996; Katz et al., 2008) reach a peak close to the base of magnetochron C13n, while the prior and first phase of the EOT transition (Step-1 of Coxall et al., 2005, and 'EOT-1' Coxall and Wilson, 2011, Zachos et al., 1996) occurs in the previous reversed polarity zone C13r. Firth et al., 2013 use 270.93 mbsf as the age tie-point for the C13r/C13n reversal boundary at Site 647. Due to the sampling limits of the paleomagnetic analysis, there is a +/- 9 m uncertainty associated with this horizon (See Supp. Table S2 in Coxall et al. 2018). Our benthic $\delta^{18}O$ sample from 269.79 mbsf falls within the zone of P-mag uncertainty. Since it has (what we interpret as) a 'pre-EOGM' $\delta^{18}O$ value (and, thus pre- C13n value), it most likely occurs within C13r. We can therefore shift the C13r/C13n reversal depth up to 265 mbsf, which is a revised estimate of the P-mag reversal position after (Firth et al., 2012) that is consistent with the P-mag constraints and $\delta^{18}O$ chemo-stratigraphy."**

**-------**

**This also reduces the depth uncertainty on the C13r/C13n reversal to ca. +/- 4 m. This revision was integrated in the Coxall et al., 2018 age model and used here.**

**While there is room for small differences in interpretation, we feel that the main position of the $\delta^{18}O$ shift, and therefore the EOT, constrained by both P-mag, and $\delta^{18}O$ chemostratigraphy, is robust. There is apparently no major change in sedimentation rate at Site 647, suggesting the section is continuous (Firth et al., 2013). Independent comparison of the $\delta^{18}O$ to the Site 647 benthic $\delta^{13}C$ is also consistent with the P-mag / $\delta^{18}O$ age framework, both locally and globally (see Coxall et al., 2018), increasing our confidence in the chemostratigraphic approach.**

**With respect to the new TEX$_{86}$ SST records therefore, importantly we have confidence in our interpretation of the relative positions of the signals discussed, i.e.**

**- the prominent cooling signal detected is pre-EOT**

**-there is no apparent temperature change straddling the acute phases of Antarctic glaciation comprising EOT-1 and EOIS (although these are not discernible in the Site 647 records directly, only by comparison to the Site 1218 $\delta^{18}O$ records).**

I am also slightly baffled by what the authors wrote in Line 231 "…no change in SST at Site 647 concurrent with the Step 1 or EOIS events…" as I do not see any SST data point in these time intervals (Fig. 2). Some clarification regarding the basis of this statement would be appreciated.

**Our point was that we do not see any permanent temperature shifts related with these events. It is true that we do not have any data point corresponding exactly with these events, but even if there was a SST change/decrease related with these events, it was short-lasting. We will clarify this.**

(2) Are TEX86- and UK'37-temperatures similar to each other?

TEX86 is a very useful proxy especially in deep-time climate reconstruction, but it is not always clear from which water depth these lipids originate and thus reflect. The authors argue that TEX86 reflects annual mean SST at their study site because they find the trend and absolute values of TEX86 temperature similar to those of the UK'37 temperatures.

- Robustness of UK'37 temperatures: Alkenones were only detected at the very end of the EOT and not for the Eocene. Does this mean that at this study site alkenone producer only started appearing during the Oligocene? What (which species) might they be? The fact that this is not the same species as modern-day precursor (E. huxleyi) on which the UK'37 calibration is based would in theory introduce some uncertainty in the UK'37 temperature estimates especially when interpreted quantitatively (absolute values). How does this uncertainty affect the interpretation of TEX86 that is based on the assumption that UK'37 temperatures are robust?

**It is well known that alkenones produced in these times are not from E. huxleyi as this species has been around for only a few 100 kyrs. Many studies have shown, however, that the ancestors show a quite similar response of the Uk37 to temperature compared to modern day producers (e.g. Villaneuva et al., 2002). Naturally, this introduces some uncertainty, similar to using extinct foraminifera or dinoflagellates to reconstruct palaeoceanographic conditions or GDGTs to reconstruct temperatures, but generally this uncertainty is considered minimal compared to other proxy errors. The fact that the alkenone producer started to appear since the Oligocene is also a common observation, i.e. alkenones really became more ubiquitous starting in the late Eocene at the end**

**of a long term cooling out of the Eocene hothouse (Brassell et al., 2014). We now briefly mention these uncertainties in the manuscript.**

- Different Oligocene trends in UK'37 and TEX86: for the interval wherein both UK'37 and TEX86 data exist, UK'37 data show a strong cooling of > 5ºC whereas TEX86 data shows little to no change in trend. Zooming in, one would see that UK'37 temperatures are higher than TEX86 temperatures during the early Oligocene but lower during 33.5–33.0 Ma. Wouldn't these different trends argue against the authors' assumption that both proxies are similar in values and trends?

**Above 180 mbsf we observe that the SST derived from Uk37' become more similar to the $TEX_{86}S$, while below that depth they are similar to the $TEX_{86}H$. This is shown in the Supplementary figure S2. However, in the interval of our interest, i.e. in the earliest Oligocene where $TEX_{86}$ and Uk37' derived SST values overlap, the temperatures are fairly identical. The slight increase in the offset between $TEX_{86}H$- and $TEX_{86}S$-derived temperatures, and the shift of the Uk37' derived SST record are however not the focal point of this paper, which is concerned with SST evolution across the EOT.**

Uk'37 calibration choice: The authors applied both Kim et al's linear regression-based calibration and Bayspar for TEX86, but only Müller et al's linear regression for UK'37. Why not also Bayspline? I think with Bayspline the abovementioned UK'37 Oligocene trend would be amplified, and further increasing the discrepancy between UK'37 and TEX86.

**The calibration proposed by Muller et al. (1998) is by far the most commonly used UK37 calibration and is also overlapping with the calibration for E. huxleyi  The calibration by Muller et al. (1998) mostly uses surface sediment data from the North Atlantic and should therefore be excellently suited to estimate UK37 SST for our core site. We will briefly discuss the reasons for choosing this calibration**.

If the authors agree that the absolute values of UK'37 temperature are uncertain and the trends in UK'37 and TEX86 are in fact different, then the interpretation of TEX86 as SST would be unsupported. As this is a very critical point for the study (see also general comment (4) on data-model comparison), some in-depth discussion is warranted to strengthen the conclusion of the study.

**See our points above. Yes, there is uncertainty with each proxy, both UK37 and $TEX_{86}$, as is the case with other temperature proxies applied in the Paleogene. We do observe an overall match in absolute values and trend but indeed, there are times of differences which may not be surprising considering the widely different ecologies of their source organisms. These uncertainties will be addressed in the revised manuscript.**

(3) Limitations of climate model

The authors mention that the 400 ppm Arctic closed simulation is not in equilibrium, and that the modern-day orbital forcing parameters were used to simulate the Eocene and Oligocene simulation. Some discussion on whether this has any bearing on the results would be helpful to convince the reader of the robustness of the conclusions (i.e. they are not affected by limitations in the model output). **The 400 ppm Arctic closed run was branch off from the more similar 800 ppm Arctic closed run. It is probably a little too warm still in the deep ocean, but the surface temperatures reach equilibrium quicker. We will show a time series of the AMOC to demonstrate the equilibrium of the circulation.**

(4) Data-model mismatch: Does TEX86 really reflect SST?

The authors discuss at length the discrepancy between proxy data (based largely on TEX86) and model output. The data-model comparison is based on the premise that TEX86 reflects SST, which hinges on whether TEX86 resembles UK'37 in absolute value and trend – the latter is not unequivocal (see my general comment (2)). Another curious observation is Figure 4 – proxy data based largely on TEX86 suggests a much flatter latitudinal gradient compared to that derived from the model output. A similar data-model discrepancy has been reported for the early Eocene – see the rather controversial study by Ho and Laepple (2016, Nature Geoscience). Ho and Laepple argue that TEX86 reflects subsurface temperatures not SST, thus an improved data-model match can be obtained when proxy data are compared to temperatures from comparable depths in the model. Might this also be the case for the EOT data-model comparison? Recently, TEX86-derived estimates at site 959 (one of the sites in the SST compilation) have been interpreted as subsurface temperatures (van der Weijst et al., ClimPast Discussion), at odds with the authors' interpretation. As the conclusion of this study hinges on interpretation of TEX86 (in other words the depth origin of sedimentary GDGTs), I invite the authors to carefully consider these points and present a more detailed analysis in the manuscript.

**There have been many studies giving extensive discussions on whether TEX$_{86}$ reflects SST and we do not want to reiterate this discussion here. From this literature it appears that the source organisms are not living in the uppermost surface waters and are mainly dominant in subsurface waters. Nevertheless, studies have also shown that the TEX$_{86}$ can reasonably predict surface temperatures and trends in deep time, in particular if there are no major differences in trends between subsurface and surface layers (the main point of the van der Weijst et al. 2021, Climate of the Past Discussion paper). Since it is not possible to predict in advance if TEX$_{86}$ is able to reconstruct surface conditions, we used an unambiguous surface temperature proxy, the UK37, to establish whether the TEX$_{86}$ can reasonably estimate past SST at our study site. We feel that, within uncertainties given and as outlined in the main document that it does. Nevertheless, we will add a more detailed discussion on this topic in the manuscript.**

Line 25: "… This step in SST values…" briefly mention how the "step" is determined. Eyeballing? Change point analysis? **We will add that this step is "visually observed" in the record. Change point analysis is a statistical technique better suited for higher resolution data sets.**

Line 112–114: Unclear how the correlation was established. Please provide more details, e.g. age tie-points or statistical technique used. **Tie-points are presented in the supplementary material.**

Line 127: UK'37 proxy was proposed by Prahl and Wakeham 1987, by modifying the UK37 proxy proposed by Brassell et al 1986. Please cite the original papers instead of later studies that applied this proxy. **Good point, we will add these references instead of Rodrigo Gamiz et al. 2015**

Line 130: Bayesian statistics-based calibration for TEX86 (Bayspar) was used, so why not also Bayspline? As the UK'37 values are rather high in this record, the choice of calibration might matter. Using Bayspline may yield higher SSTs with a larger magnitude of change compared to those obtained using the Müller et al 1998 calibration. **See our comments above. The Muller et al. calibration is by far the most commonly used UK37 calibration and mostly uses surface sediment data from the North Atlantic and should therefore be excellently suited to estimate UK37 SST for our core site.**

Line 181–183: See my general comment (2).

Line 183–187: I find this argument a little confusing. Based on the results of a culture experiment, Qin et al. proposed that archaea may change their GDGT distribution in response to changing oxygen concentration in their living environment. Since the authors interpret TEX86 temperature as SSTs, the implicit assumption is that the sedimentary lipids must necessarily come from planktonic archaea living in the mixed layer. It thus follows that it would be more logical to assess the O2 concentration in the habitat of the archaea in the upper water column rather than that of the depositional environment of the lipids after cell lysis.

**Qin et al (2015) argued that the predominant times when $O_2$ levels would affect the $TEX_{86}$ is at times of low oxygen concentrations in large parts of the water column, i.e. oceanic oxic events and oxygen minimum zones. Therefore, we discuss a potential evidence for low oxygen conditions at our core site. In fact, it is extremely rare to encounter low oxygen conditions in the surface waters in the last few 100 million years. Note that we do not argue here that the archaea are living in the surface waters: plenty of studies have shown that they predominantly come from the subsurface waters. However, organic matter transport may bias the signal towards the upper part of the water column.**

Line 189–190: On average, <10% of the organic matter that is produced in the photic zone ends up in marine sediments at the seafloor. All organic matter in the marine system is subject to degradation, GDGTs included. As for no sharp increase in BIT – this would only be apparent if there is a large change in O2 in sediments, e.g. in turbidite sequences. But the fact that we do not see it does not mean there is no degradation of OM. Please rephrase the sentence to improve clarity. **We will rephrase it to make it more clear**

Line 198 & 207–208: See my general comment (3). **Addressed in reply to comment (3)**

Line 216–219: See my general comment (2). **This is addressed in our reply to general comment 2**

Line 231–232: See my general comment (1). **This is addressed in our reply to general comment 1**

Line 242: "…decreased more permanently." The usage of "permanent" here is a bit confusing. Please reword or clarify. Would "substantial" or "prolonged" work better in this context? Also correct it throughout the manuscript. **This will be clarified and made consistent throughout the manuscript.**

Line 272–276: See my general comment (2). Just because the global core-top data fits better with annual mean SST does not mean that the alkenones at site 647 reflect the annual mean too. Previous studies have reported a better fit between North Atlantic core-top UK'37 and seasonal SST (e.g. Tierney and Tingley, 2018 Paleoceanography and Paleoclimatology). Also, might it be an idea to compare the TEX86 temperatures to the summer SST in the model? **We will add a comparison of the proxy data with summer SST from the model simulations to the supplementary material.**

Section 5.2 and 5.3: Spotted numerous typos. Please proof-read before resubmission. **This will be corrected**

Line 318–320: The temperature maximum and minimum mentioned here is based on one or a few data points. Are these statements supported by the data presented, given the uncertainty in age model and proxy noise? **We will address this issue in the new**

**version of the manuscript**

Line 323–324: It IS really based on only one data point. Please provide more robust evidence or rephrase the sentence. **We will in the new revision clearly admit this and take out the word "arguably" in the sentence "the variability does not rely on individual data points except arguably for the minimum at 35.7".**

I think Section 5.1 and 5.3 can be merged, or at least 5.3 follows 5.1, before the discussion on data-model comparison. **All sections in Section 5 have some form of data-model comparison. Section 5.1 is all about the realism or not of the general Atlantic absolute temperatures, section 5.2 moves to general Atlantic temperature variability across the EOT, and section 5.3 moves then more specifically to the variability of the local site temperature. We will improve the logical outlay by updating the headings of the sections to clarify how they logically follow from each other. Specifically, the section will be**

**5.1 Absolute SST values in the NA**

**5.2 SST change across the EOT in the NA**

**5.3 SST variability across the EOT at site 647.**

Line 332–336: Is it possible that the EOT cooling at site 647 is also caused by a long-term shift in the gyre boundary? **This is what we are suggesting: as marked by the SST drop, the site area remained under the influence of the polar gyre**

Figure 3: It is difficult to tell apart the colors in panel B and to match the lines to the site locations. Perhaps try a different color palette? It might also be helpful to add a legend. **We will change to a different colour palette, as suggested**

---

## Referee Report (RR1)

**Referee comment on 'Sea surface temperature evolution of the North Atlantic Ocean across the Eocene-Oligocene Transition' by Śliwińska et al.**

In light of previous comments, the authors have done a good job addressing most issues while sticking the important information. It is now clear that this paper focusses on the proxy record of North Atlantic SST, trying to explain some of their observations to the results of climate models but leaving a more in-depth discussion of the latter outside of its scope.
Language has been improved, with some errors remaining mostly due to textual changes made that should be resolved in a final sweep.

The main issue remains with readability/clarity of some of the figures, which I feel are not addressed properly. Some of the choices were explained in the authors' response, but little to no adjustments were made;

- Figure 4: there is too much information stacked on top of each other, using different colours, boxes and lines. I don't see much added value showing present-day SSTs (unless this serves as e.g. a model reference), yet using this over an anomaly with respect to PD uses about half of the figure. As it is now, it is also not clear to me what the main message is to the reader; the difference between the time intervals, with respect to the present, or rather which model simulation performs best?
- Figure 5: is a nice addition, but many of the trends are obscured by the large shifts during the asynchronous coupling phase. Consider leaving out the first ~3300 years, or splitting the figures into 2 parts adjusting the vertical axes. For consistency, it would be nice to colour-match the simulations throughout the different figures.
- Figure 6 (was 5): This is indeed a useful figure, but it does not serve its intended purpose well because of the very subtle colour scale and large range. In its current format, a certain proxy SST can easily be off by 4C or more and barely be visible. I suggest using a different colourmap and/or narrower temperature range such that the differences between both proxy and model SSTs are clear for the different scenarios.
- Figure 8 (was 7): I agree that the BSF is a very useful measure and that it should be depth averaged. Maybe my point was a bit unclear and was rather meant to take care in interpreting what is shown. The discussion of this figure mainly aims to explain part of the SST changes through current changes and the extent of SP/ST gyres. As the gyre contributions are likely quite week at this point in the North Atlantic, the differences between both figures are probably mostly AMOC transports. Therefore, the link between SSTs and changes in depth-integrated transports as well as gyre extent is therefore in my mind not easily made based on the results shown.

**Minor comments:**

- L280 and following: SST proxies are still referred to as observations here.
- L360: This is confusing; insufficient polar amplification would mean too cool rather than too warm polar temperatures? Figure 5 also shows that the simulations are underestimating rather than overestimating high latitude temperatures, in contrast to what is mentioned here. Additionaly, 'polar' temperature is somewhat of an unlucky term, as all of the proxies considered are near or equatorward of the Arctic Circle. Use 'middle/high latitude' instead?
- Section 5.2: The authors suggested to add further information on the meridional overturning stream functions and SST of deep water formation regions. Much of this section was adjusted and a figure showing AMOC timeseries has been added. If there is any additional information and/or analysis in the supplement, there is no mention or reference here.

---

## Author Response (AR2)

**Comments from the authors:**
- We have found the comments from both reviewers very useful. We addressed them below.
- In addition to this, we have updated some figure captions. We have also updated and increased the resolution of some of the figures in the Supplementary file.
- We can move the raw data from the excel file into the PANGEA database when the manuscript is accepted
- The DOI for the model data will be activated once the manuscript is accepted.
- The reference list of the Supplement file is adjusted to the journal standard, as requested.

**Replies to the reviewers:**

**Report 1 (Baatsen)**

Referee comment on 'Sea surface temperature evolution of the North Atlantic Ocean across the Eocene-Oligocene Transition' by Śliwińska et al. In light of previous comments, the authors have done a good job addressing most issues while sticking the important information. It is now clear that this paper focusses on the proxy record of North Atlantic SST, trying to explain some of their observations to the results of climate models but leaving a more in-depth discussion of the latter outside of its scope.

Language has been improved, with some errors remaining mostly due to textual changes made that should be resolved in a final sweep. The main issue remains with readability/clarity of some of the figures, which I feel are not addressed properly. Some of the choices were explained in the authors' response, but little to no adjustments were made;

- Figure 4: there is too much information stacked on top of each other, using different colours, boxes and lines. I don't see much added value showing present-day SSTs (unless this serves as e.g. a model reference), yet using this over an anomaly with respect to PD uses about half of the figure. As it is now, it is also not clear to me what the main message is to the reader; the difference between the time intervals, with respect to the present, or rather which model simulation performs best? **We have improved the figure as follows:**
**- the color palette match with the model output in figure 5 - it is a very good point regarding better colour match between figures!**
**- the main message is to compare the various model outputs with the proxy data – for this purpose it makes more sense to keep both: the model outputs and the proxy-data records.**
**- the modern SST are shown as a reference, so we would like to keep it. However, we have toned it down, by reducing it to a single line, so it is not dominating the figure.**

- Figure 5: is a nice addition, but many of the trends are obscured by the large shifts during the asynchronous coupling phase. Consider leaving out the first ~3300 years, or splitting the figures into 2 parts adjusting the vertical axes. For consistency, it would be nice to colour-match the simulations throughout the different figures. **We have left out the first 3000 years and adjusted the size of each of the time-series. The color code for each model output is applied also in Fig. 4 for better synergy between figures.**

- Figure 6 (was 5): This is indeed a useful figure, but it does not serve its intended purpose well because of the very subtle colour scale and large range. In its current format, a certain proxy SST can easily be off by 4C or more and barely be visible. I suggest using a different colourmap and/or narrower temperature range such that the differences between both proxy and model SSTs are clear for the different scenarios. **This is a valid point. We have applied a colour-blind friendly palette, which is rather limiting. Other colour palettes did not improve the figure significantly and were not colour-blind friendly. Therefore, instead we added the SST data points for each site (pre- and post-34.5 Ma) on top of the temperature-colour scale. We believe that it makes the figure easier to read.**

- Figure 8 (was 7): I agree that the BSF is a very useful measure and that it should be depth averaged. Maybe my point was a bit unclear and was rather meant to take care in interpreting what is shown. The discussion of this figure mainly aims to explain part of the SST changes through current changes and the extent of SP/ST gyres. As the gyre contributions are likely quite weak at this point in the North Atlantic, the differences between both figures are probably mostly AMOC transports. Therefore, the link between SSTs and changes in depth-integrated transports as well as gyre extent is therefore in my mind not easily made based on the results shown.

**We agree that the SST changes are not strictly wind-driven gyre changes because of the contribution to the AMOC. However, the SST increase when closing the Arctic has a very sharp front that sits right on the core site, and it corresponds also the boundary of the barotropic streamfunction. It makes therefore sense that the SST changes are related to the circulation changes (as there is no other obvious explanation for such a sharp SST front in the SST anomaly (see Fig7a, b, e). We now make it clear that this is not strictly related to the wind-driven gyre. In the discussion of Figure 8, we now note explicitly that: " The barotropic streamfunction in the model, a combination of the wind-driven gyre transport and the meridional overturning streamfunction, suggests ... ".**
**We further say: " with the streamfunction-derived Subtropical gyre" instead of just Subtropical gyre" to clarify it is not strictly the same.**
**And finally, we write that: "The position and strength of the gyres are likely model dependent and is here affected by the AMOC,"**
**These changes should clarify our point that the ocean circulation in the region (from winds and AMOC) is highly dynamic so could easily influence the SST there.**

Minor comments:
- L280 and following: SST proxies are still referred to as observations here. **It is corrected now**

- L360: This is confusing; insufficient polar amplification would mean too cool rather than too warm polar temperatures? Figure 5 also shows that the simulations are underestimating rather than overestimating high latitude temperatures, in contrast to what is mentioned here. Additionally, 'polar' temperature is somewhat of an unlucky term, as all of the proxies considered are near or equatorward of the Arctic Circle. Use 'middle/high latitude' instead?
**Thanks for catching this error. This should be "too cold polar temperatures" of course. Also, we have changed throughout the manuscripts the description of the "polar" temperatures to high latitude temperature as we agree this is more descriptive of where these proxy locations are.**

- Section 5.2: The authors suggested to add further information on the meridional overturning stream functions and SST of deep water formation regions. Much of this section was adjusted and a figure showing AMOC timeseries has been added. If there is any additional information and/or analysis in the supplement, there is no mention or reference here.
**We added the meridional overturning stream functions series and deep ocean temperatures that we suggested to add in Figure 5. This concludes the additional analysis - as we explained, we want to avoid turning this too much into a modelling paper so only addressed the aspect directly related to the data.**

**Reviewer 2:**

The authors have made substantial changes to address the concerns raised by reviewers. These changes include adding more detail about the uncertainty/limitations of both proxy data and model simulation, as well as adding and revising figures to improve clarity. I also appreciate that some paleoclimatic interpretation has been toned down or caveated accordingly. The authors have added some new text outlining the limitations of their model runs which were not in equilibrium. This seems like a critical issue but I do not have the expertise to assess. Overall I do find the manuscript much improved. However, there are still some technical issues and unclear reasoning that should be resolved. I also think that some minor restructuring of the manuscript might help further improve its

clarity.

[Line numbers as in the tracked changes version of the manuscript]

Line 40–44: These two sentences give off the impression that it is not possible to draw any firm conclusions from the results because the data reported are uncertain and model runs were not in equilibrium. This might lead to some readers thinking what is the point of this paper. I think the results presented are more valuable that that implied by these two sentences. Perhaps rephrase it to something more positive along the line of "future work"?
**We agree and have now rephrased the abstract ending to conclude more positively as follows: " We conclude by highlighting remaining uncertainty in various aspects of proxy and modelling work which could be improved to shed further light on the processes responsible for the cooling trends identified here prior to and across the EOT in the high latitude North Atlantic."**

Related to this, a dedicated section on future work / suggestions on what else to do to improve our understanding of temperature evolution across EOT might improve the structure and clarity of discussion. I would also shorten the conclusions by moving all the suggestions / future work to this proposed new discussion section.
**The uncertainties related to the conclusions are a result of data scarcity and uncertainty, as well as model uncertainty and the fact that these simulations explore only very few of the possible EOT scenarios. Any model or data study is likely to help with, but none to solve alone the question of the EOT. We have thus concluded with a general recommendation that more proxy SST data and modelling results are needed. However, we have now added at the end the suggestion of a formal EOT model intercomparison project.**

Line 164: "where 37:3 stands for methyl ketone and 37:2 for alkenone". Both are alkenones, which are ketones. The numbers refer to the number of carbon atom and double bonds in the molecule, respectively. Please revise accordingly. Also the notation [C37:2] is more common that 37:2.
**We have corrected the text as suggested.**

Line 170: "…has been present only for 270 kyr." 270 kyr during which time interval? Suggested change "the past 270 kyr". Also a reference is needed here.
**We have corrected the text as suggested.**

Line 175: "…overlapping with the calibration used for E. huxleyi." Specify which calibration; several calibrations derived from E. huxleyi have been reported, some of which are very different from the Müller calibration. Would be also helpful to briefly explain why similarity with a culture calibration confirms the robustness of the core-top calibration.
**We have added a relevant reference. See also our response to the comment below. We think that any further explanations discussing regarding similarity with a culture calibration are out of the scope of this paper.**

Line 175–177: The Müller calibration is based on data from the global ocean, so it is not true that it "mostly uses surface sediment data from the North Atlantic Ocean…". That would be the Rosell-Mele et al 1995 GCA Atlantic calibration.
**This is a valid point, which we considered. We have rephrased the sentence as follows the calibration of Müller et al., (1998) is near identical to the culture-based calibration used for *E. huxleyi* by Prahl et al. (1988) and is the most commonly used calibration for the UK37'-derived SST calculation of the late Paleogene to Neogene strata in the northern high to mid latitudes (see e.g. Liu et al., 2009; Herbert et al., 2020; Weller and Stein, 2008)."**

Line 225: Fig S2 shows temperature records based on different calibrations, not calibrations (ie regressions) themselves. Please rephrase. **Good point. It is corrected now.**

Line 231–232: Unclear reasoning. Just because proxies are based on organisms with different

ecological preferences does not mean that they are associated with uncertainty. Do the authors mean inter-proxy comparison is associated with uncertainty? **We mean that each of the proxy not perfectly reflecting annual mean SST (i.e. 0-30 mixed layer) but is generally reflecting different seasons and different depths, and therefore carry different uncertainties with respect to reconstruction annual mean SST. The sentence is rephrased to: "These two proxies are based on organisms with different ecological preferences and thus may reconstruct temperatures of different seasons and in particular depth compared to each other." And in the given context we believe that the meaning is now clearer.**

Line 232–234: As I commented in the previous round of review, the reported UK'37 and TEX86 records (the entire time series) are really not that similar in trends nor absolute values; the UK'37 record shows a strong cooling trend while TEX86 record shows very little change. The match is only true for the early Oligocene. I would therefore be specific with the time interval during which proxies match. **We have stated that the similarity in trends is in the given interval "(covered by the interval from ~240 to ~190 mbsf), line 221). The magnitude of cooling at ~183 m depth is different between the proxies, but it is observed in both proxy records. Also, as shown on Fig. S2B the trends are fairly similar. As stated above, this is to be expected from proxies which are from organisms with different ecological preferences.**

Another piece of information regarding surface vs subsurface origin of TEX86 signal is the GDGT-2/GDGT-3 ratio (Taylor et al 2013 GPC), which is used by some workers to indicate deep-water archaeal contribution to sedimentary GDGTs.
**The GDGT-2/GDGT-3 ratio was calculated for all samples and is shown in our supplementary file. The ratio is low (< 5) and rather stable, suggesting no major subsurface contribution to sedimentary GDGTs.**

Line 280: I would remove "Observations of". Strictly speaking proxy-derived SST estimates are not really "temperature observations". **Good point, we have removed word "observation" from lines 285 and 286, and changed to: "4. Proxy-derived sea surface temperature" and "4.2. Sea surface temperature in the North Atlantic across the EOT".**

Line 287–288: Unclear what is meant by this sentence. Some elaboration would be helpful. **We have rephrased the sentence to make our intentions clearer.**

Line 320: "Proxy-derived" instead of "Observed". **It is corrected**

Line 408: "latter" not "letter". **It is corrected**

---

## Author Response (AR3)

**Dear Bjørg Risebrobakken**

**Thank you for your valuable comments. Our replies are marked below in blue**

- We have added a new figure in the Supplementary Information file (Fig. S5 – scatter plot comparison of model annual mean sea surface temperatures with the proxy data)
- Few new relevant references are added (highlighted in red)
- The dataset with all new data will be included in the FAIR database (dataverse.geus.dk). When referring to the dataset in the manuscript text, we write "Supplements", but I am not entirely certain that this is the proper way to do. Once I have the doi for the dataset, it will be possible to include it in the reference list.

**Comments to the author**:
Dear Kasia Sliwiska and co-authors,

Thanks for sending the revised manuscript and your responses to the reviewers' comments. I have now gone through the manuscript and the responses. There are still some issues that I would like you to address before accepting the paper (see comments bellow). I will invite you to submit your new version in a clean and a track changes version, as well as your replies to my comments.

Best regards,
Bjørg Risebrobakken
Editor, Climate of the Past

All my comments refer to the track changes version of your resubmitted manuscript.

One thing that stoke me when reading the paper again is that it is unclear exactly what part of your record you refer to at different times, and which of your datasets you refer to. Your data cover 38-26.5 Ma, however, mostly you only discuss the interval 37-32 Ma. Hence, a lot of the new data is not really discussed in detail.

**This is a valid point. We focused on the interval between 37 and 32 Ma because this was the maximum time span with SST data coverage across the EOT. Secondly, the age model for the interval 32 to 26.5 Ma is rather poorly constrained. But we recognise that this should be addressed better, so we have now modified a part of section 4.1 as follows** "Overall, both paleothermometers suggest Oligocene SST (interval from 34.4 Ma to ~26.5 Ma) below 26°C (Figs 2 and S2), with two temperature minima. However, with the existing uncertainties in the age model for this interval (i.e. depth from 190 to 130 mbsf; Firth et al., 2013) it is challenging to link the SST minima with the cooling episodes from the Oligocene (e.g. Wade and Pälike, 2004). This could potentially be improved by a more detailed analysis of dinocysts (e.g. Śliwińska et al., 2010; Śliwińska, 2019), but it is outside the scope of the present study. Notably, at the older SST minimum (depth ca. 183 mbsf, Fig. S2) $U_{37}^{K'}$- derived SST becomes significantly colder than $TEX_{86}$-derived SST. Potentially this may be due to the fact that the surface conditions, reflected by the $U_{37}^{K'}$-, changed more substantially than subsurface temperatures,

which will affect TEX$_{86}$-derived to a larger extend. Alternatively, it could indicate that there were shifts in seasonal impacts on the proxies."

**We have now also addressed the absolute ages for the Oligocene better in section 2 (lines 127-132):** "The absolute ages for the studied succession are calculated up to the depth of 214.19 mbsf, where the highest occurrence of *Reticulofenestra umbilicus* (with diameter >14 µm) is observed, which provides an absolute age of 32.02 Ma at that depth (Firth et al., 2013). The uppermost part of the studied succession belongs to the NP24 (Firth, 1989) and the normal polarity magnetochron (Firth et al., 2013), suggesting that it is probably not younger than 26.5 Ma.".

**Furthermore, the SST evolution across the EOT is complex enough, so we did not want to diffuse the focus of our discussion too much.**

When you zoom into the 37-32 ma interval, only a fraction of your alkenone data is included **The alkenones appear only around 33 Ma, as we mention in lines 280-282**

When you compare your data to other North Atlantic records you do only for a selected time interval (37-32 Ma), not the full reconstructed time interval. Why? **Due to all the data and model uncertainty, it is already challenging to convincingly couple the signals found to processes. So, a lot of carefully writing was needed not to mislead the reader. If we include more and even more uncertain data, the manuscript could become dominated with what we don't know and loose its power**

I would very much like you to g carefully through the text and specify what proxy and time interval you refer to at different times. Some places you use Eocene and Oligocene, but it's not clear if you refer to the 37-32 Ma interval or your full records. Similar when you use pre and post 43.5 Ma – does this always refer to 37-34.5 and 34.5-32 Ma? Make sure that it is clear throughout the paper what time interval you refer to. **We have followed this suggestion and carefully checked across the entire text, see (track changes version) lines 287, 298-299, 343, 368, 463, and figure captions.**

The first reviewer asked you to do a careful language check of the text. There are still quite some places where the sentences can be hard to follow, or where words should be deleted or added (e.g. line 324 … the where the…); as reviewer 1 mentioned this may have emerged through the revisions, but should still be taken care of. **The language is checked now**

Line 174: suggested rephrasing: .. and is commonly used to estimate Uk`37-derived SSTs of the… **rephrased as suggested**

Line 234: I agree with reviewer 2 that it's not clear from the way it's written that this statement refers to the early Oligocene only (even if you have the statement earlier in the same paragraph). Suggested rephrasing: … the similarity of both records during the earliest Oligocene (x-y Ma) suggests that at this time the temperatures recorded… **This is a valid point, and the suggestion is followed. However, we are not able to provide absolute ages as suggested, since it is not possible in this depth interval. We have addressed this issue in section 2.**

Line 285/Fig. S1: The statement that the data are in the same range; yes, it is true for the early Oligocene, but not above ca 270(?) m. However, it's not true for the younger interval/when you look at your full record. In the younger end, your alkenone SSTs are much colder than the TEX SSTs. There is a clear transition in the alkenone data; maybe I missed it, but I cannot see that this divergence between the proxies is discussed anywhere **This is addressed by our new text:** "Overall, both paleothermometers suggest Oligocene SST (interval from ~34 Ma to ~26.5 Ma) below 26°C (Figs 2 and S2), with two temperature minima. However, with the existing uncertainties in the age model for this interval (i.e. depth from 190 to 130 mbsf; Firth et al., 2013) it is challenging to link the SST minima with the cooling episodes from the Oligocene (e.g. Wade and Pälike, 2004). This could potentially be improved by a detailed analysis of dinocysts (e.g. Śliwińska et al., 2010; Śliwińska, 2019; Śliwińska and Heilmann-Clausen, 2011), but it is outside the scope of the present study. Notably, at the older SST minimum (depth ca. 183 mbsf, Fig. S2) $U_{37}^{K'}$- derived SST becomes significantly colder than TEX86-derived SST. Potentially, this may be because the surface conditions, reflected by the $U_{37}^{K'}$, changed more substantially than subsurface temperatures, which will affect TEX86-derived to a larger extend. Alternatively, it could indicate that there were shifts in seasonal impacts on the proxies." **Since the data on this is limited at the moment, we would not like to dive into this in details. However, we think that it is important to show all the data span we have produced and hope that when more data appears we could have a better idea how to explain this.**

L288: Cn you specify the ages off the intervals you compare? The statement that its <25°C after the transition is true for TEX, but not really for the early Oligocene alkenone data. If you take into account the uncertainty (that you use to say that the proxies show the same (S1) both proxies are potentially above 25°. **This is a good point. We have specified these, by modifying the text in lines 297-299 as follows:** "Overall, $TEX_{86}^H$ - derived SST shows a distinctive cooling step of ~3-4 ºC at Site 647, when comparing the warmer Eocene (SST between 29 °C and 25.5 °C, interval from ~38 to 35.5 Ma) with the colder Oligocene (SST below 25 °C, interval from 33.5 to ~26 Ma) "

Line 289: You refer to Table 1 - where is this table? I cannot find it anywhere. **This is an error on our side, the data are shown in the excel file as one of the supplementary information files. All references to Tables are checked and corrected now.**

Line 322: Be clear that pre and post 34.5 Ma does not include your full record. Rather be specific and use the exact time intervals you refer to. Also, it is not totally clear why you set the change point at 34.5 Ma **We explained it in line 334-335:** "The threshold of 34.5 Ma is chosen, because that is where the clear shift towards colder temperatures in Site 647 is recorded. " Please be more specific on the reasoning behind this choice. Is it the midpoint of the transition as seen in your data? . **We have modified lines 343-346 as follows:** "Here we compare the late Eocene (37 to 34.5 Ma) and early Oligocene (34.5 to 32 Ma) SST at the five North Atlantic core sites to the four combinations of an open and closed Arctic, and 400 and 800 ppm atmospheric CO2 concentrations as described in Hutchinson et al. (2018, 2019). The selected time frame from 37 to 32 Ma covers the most complete data-derived SST evolution from all selected sites (Fig. 3). "

IF so, how do you define the start/end? **We have focused on the 5 million year bracket around the EOT, where the SST data coverage from all sites is the most complete.** Or is it the the starting/end point (does not look like it)? Would it make a

difference to your results if you looked at the pre and post transition intervals instead of including the transition in the SST estimates that you compare with the model results? **Skipping the SST transition interval in our data-driven ΔSST would probably show slightly larger Eocene SST in site 647, but we would lose many data points in site 1404, and few in some other sites. All the existing records, compared to our 647 SST record, are of very low resolution, so we had to compromise in order to get the most optimal view on the SST transition.** In your figures you visualize EOT, however, your transition point is not set at the same time. Please visualize your transition point in the figures as well as the general EOT. **The cooling/transition point coincides with the Late Eocene cooling. We have the Late Eocene event marked on all figures with the SST data, including the absolute ages. However, we now made it clearer (line 302) that the cooling relates with the Late Eocene cooling, so it should be easier to follow for readers.**

You refer to Table S2 here, however, there is no table in your supplement (neither a Table S1 nor S2). **It was error on our side. The data referred here to Table S2 are now incorporated in Figure 6.**

Line 430: Suggested rephrasing: The position and strength of the gyres, as well as the strength of the AMOC, is model dependent … **it is rephrased**

Line 447: before the 34.5 Ma cooling step? **Yes, it is corrected now**

Line 573: Is there room for being more specific on your recommendations here? SST data in general? Preferentially multi proxy reconstructions from the same sites, or preferentially larger spatial spread from one proxy? Pros and cons with both approaches? What about other proxies that can help characterise the climate system before, during and after the transition? **Our work focus on the marine records, so we have added more specific recommendations in lines 487-495 regarding marine proxies:** "This calls for more proxy records SST data to increase the spatial coverage and resolution of regional temperature trends across the North Atlantic in order to identify possible thermohaline fingerprints of the AMOC start-up at the EOT. For areas located south from site 647 and Kysing-4 this could include construction of east-west Atlantic surface and deep water δ18O and temperature gradients using multiple palaeotemperature proxy methods (e.g. clumped isotopes, foraminiferal Mg/Ca, or TEX86). For the higher northern latitudes, where calcareous microfossils fossils are very limited in this time interval, this could include higher resolution SST proxy reconstruction based on TEX86 and/or $U37K'$. Despite the existing hiatuses at the EOB interval in the North Atlantic region, increasing sampling resolution in the existing sites in the interval from 37 to 32 Ma would be beneficial. ". **The added information reflects well in the state of the art outlined in the Introduction (lines 69-75)**

Figure 4: Please be more specific in the caption: 1) zonal average - global of for the Atlantic sector? 2) Specify that WOA is shown as a reference, and for what it is considered a reference. 3) In the captions you use >34.5 and <34.5, while in the figure you are more specific (37-34.5 and 34.5-32. Please be consistent and use the exact time interval also in the caption. 4) specify that its TEX data presented - or is the mean calculated based on a mix of proxies for the younger interval? It should be clearly

specified what is included where. Also be clear on why you make the choice to integrate the proxies if that is what you have done. **We have clarified these issues in the figure caption and other relevant places (e.g. caption to Fig 6 and 7).**

Figure 6: I agree that the added changes contribute to visualise the differences between the reconstructed SST of the pre and post 34.5 Ma intervals. However, it does not solve the problem of seeing clearly how these temperatures relate to the modelled temperatures of the different scenarios. Maybe it is one option to show the modelled SSTs at the core sites in a similar way as you do now with the proxy data, for each scenario? Make an additional panel or add to the legend one line where you place the relevant information where it should be relative to the legend for each model scenario as well as the now added pre/post symbols. Would also be a cleaner presentation if all the squares were at one line, the circles at one line, and then similar for each model scenario. **We agree with this comment, but when these suggestions are applied, the figure becomes messier, in our opinion. For more detailed proxy and model data the reader can follow Fig.4, where all site-specific simulated SST are shown as triangles. We have added a reference in the figure caption, the raw data are included in the excel spreadsheet (online dataset) and a new figure S5.**

Figure 7: TEX data only? Please specify. **It is specified now**

Figure S4: For Figure 6 you argue for the use off a colour scheme fitted for colour blindness. Why do you not follow the same line of logic here? **We have now applied the same colour palette as in Figure 6.**

**My best regards**

**Kasia K. Śliwińska**

---

## Author Response (AR4)

Dear Bjørg Risebrobakken,

Thank you for your comment regarding the figure, it was a good point, indeed!

I have added a new line in the acknowledgements (text file)

I have made two minor aesthetic corrections of the other existing figures:

Figure 1: I have made the "Norwegian-Greenland Sea" more visible.

Figure 2: I have added brackets to "a" and "b" to make it more in line with remining figures and aligned the upper text line in the Figure 2

I hope that this is OK.

My best regards

Kasia K Sliwinska